# Unexpected assembly machinery for 4(3*H*)-quinazolinone scaffold synthesis

Xi-Wei Chen[1], Li Rao[1], Jia-Li Chen[1] & Yi Zou [1]✉

4(3*H*)-quinazolinone is the core scaffold in more than 200 natural alkaloids and numerous drugs. Many chemosynthetic methodologies have been developed to generate it; however, investigation of its native enzymatic formation mechanism in fungi has been largely limited to fumiquinazolines, where the two nitrogen atoms come from anthranilate (N-1) and the α-$NH_2$ of amino acids (N-3). Here, via biochemical investigation of the chrysogine pathway, unexpected assembly machinery for 4(3*H*)-quinazolinone is unveiled, which involves a fungal two-module nonribosomal peptide synthase ftChyA with an unusual terminal condensation domain catalysing tripeptide formation; reveals that N-3 originates from the inorganic ammonium ions or the amide of L-Gln; demonstrates an unusual α-ketoglutarate-dependent dioxygenase ftChyM catalysis of the C-N bond oxidative cleavage of a tripeptide to form a dipeptide. Our study uncovers a unique release and tailoring mechanism for nonribosomal peptides and an alternative route for the synthesis of 4(3*H*)-quinazolinone scaffolds.

Quinazolinone, a form of oxidized quinazoline, is one of the most important scaffolds found in numerous complex metabolites produced by plants, bacteria and fungi[1,2]. Quinazolinones can be classified into three types according to the position of the carbonyl group: 4(3*H*)-quinazolinone, 2(1*H*)-quinazolinone and 2,4(1*H*,3*H*)-quinazolinone (Fig. 1a). Among them, 4(3*H*)-quinazolinone is the core scaffold in more than 200 naturally occurring alkaloids[3]. In medicinal chemistry, 4(3*H*)-quinazolinone acts as the predominant functional group in various first-line antitumor or sedative agents, such as raltitrexed, idelalisib and methaqualone (Fig. 1b), or as the ubiquitous structural scaffold in numerous marketed drugs (Supplementary Fig. 1) that possess a multitude of other pharmacological activities, such as anti-malarial, anti-inflammatory, anti-HIV, antifungal and antidiabetic properties[4,5]. Moreover, recent utilization of the 4(3*H*)-quinazolinone scaffold for the design and synthesis of antibacterial agents, especially for those combating multidrug resistant microorganisms, has greatly accelerated the development efficiency of novel antibacterial drug leads[6,7].

Due to the broad pharmaceutical applications of 4(3*H*)-quinazolinone, chemists have developed a series of methodologies to synthesize this valuable scaffold and its derivatives[8,9]. However, in the field of biosynthesis, the native enzymatic formation mechanism of 4(3*H*)-quinazolinone has been largely limited to the fungal fumiquinazoline family of peptidyl alkaloids[10]. As shown in Fig. 1c, biochemical characterization of the canonical three-module nonribosomal peptide synthase (NRPS) TqaA shows that cyclization of the linear anthranilate-D-tryptophan-L-alanyl tripeptide to generate fumiquinazoline F (FQF) is catalysed by the terminal condensation domain ($C_T$ domain) in a thiolation domain-dependent fashion[11]. The $C_T$ domain likely first catalyses N-1-C-10 bond closure to form the proposed macrocyclic intermediate. Subsequently, the 4(3*H*)-quinazolinone scaffold is formed via the spontaneous attack of the C-10 carbonyl group by N-5[12]. Additionally, a non-native pathway for 4(3*H*)-quinazolinone generation was observed by the extra function of the $Fe^{2+}$/α-ketoglutarate-dependent dioxygenase (α-KGD) AsqJ on benzo[1.4]diazepine-2,5-dione substrates (Fig. 1d)[13]. Notably, these pathways are the only currently identified and biochemically confirmed assembly machineries for the synthesis of the 4(3*H*)-quinazolinone scaffold in fungi, where N-1 originates from anthranilate (Ant), and N-3 comes from the α-$NH_2$ of amino acids or their analogues.

In addition to the 4(3*H*)-quinazolinone peptidyl alkaloids shown in Fig. 1c and Supplementary Figs. 2, 3, another unique example is chrysogine (**1**; Fig. 2), a yellow pigment produced by several fungal genera, including *Penicillium* sp., *Aspergillus* sp. and *Fusarium* sp[14–16]. In

[1]College of Pharmaceutical Sciences, Southwest University, Chongqing 400715, P. R. China. ✉e-mail: zouyi31@swu.edu.cn

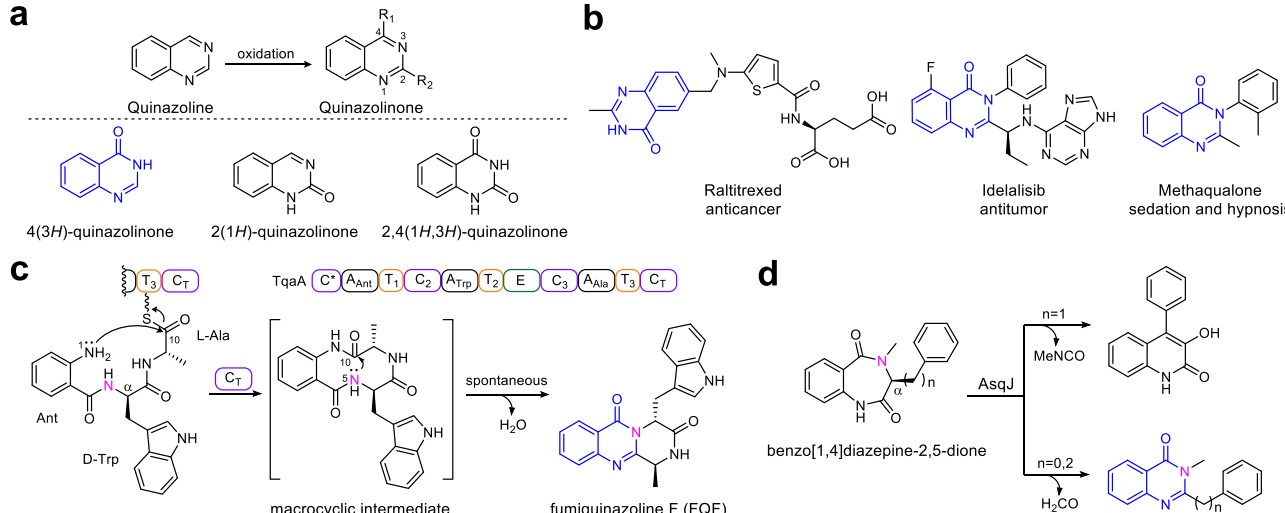

**Fig. 1 | Representative 4(3H)-quinazolinone scaffold-containing drugs and their biosynthetic machinery in natural products. a** Three quinazolinone scaffolds and **b** the best-selling drugs containing a 4(3H)-quinazolinone scaffold. **c** 4(3H)-quinazolinone scaffold formation is catalysed by the NRPS $C_T$ domain and **d** α-KGD-mediated rearrangement.

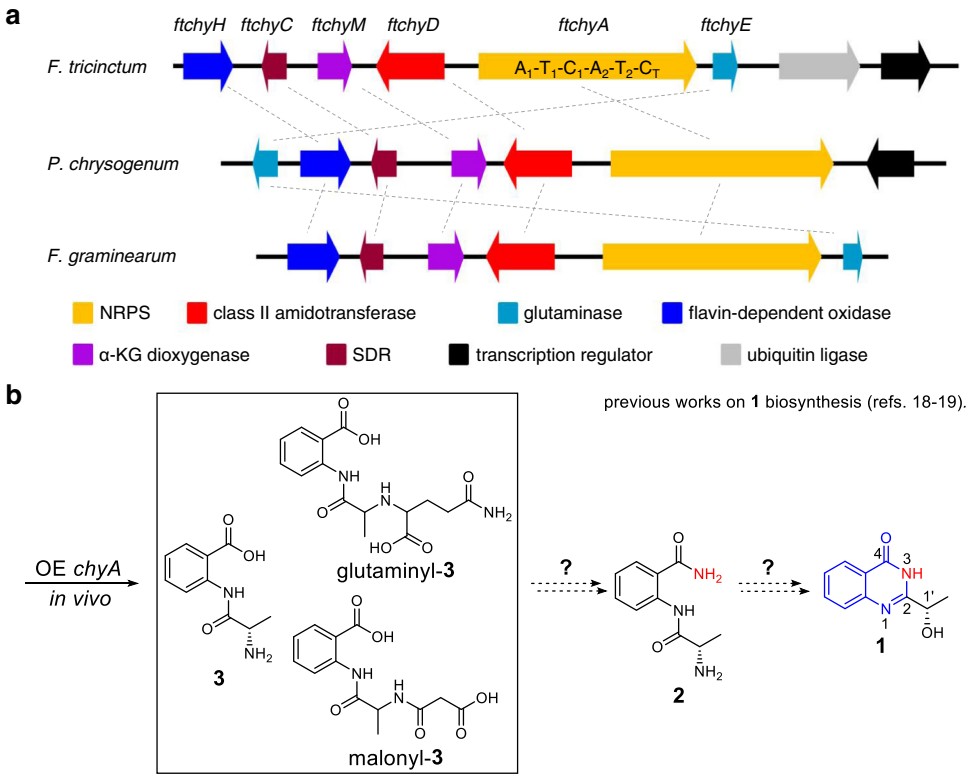

**Fig. 2 | Gene clusters and previously proposed pathway for the synthesis of 1. a** Three homologue clusters for the synthesis of **1** in different fungi. **b** In vivo gene overexpression and knockout experiments in *P. chrysogenum* and *F. graminearum* suggested that **3** or its derivatives and **2** are the possible precursors of **1**.

*Fusarium* species, **1** acts as a predominate mycotoxin during infection of both food crops (wheat and barley) and cash crops (apples)[17]. **1** is the simplest 4(3H)-quinazolinone-containing natural product; however, in contrast to fumiquinazoline biosynthesis (Fig. 1c), the N-3 in **1** does not seem to be directly derived from the α-NH₂ of amino acids, thereby indicating that an unrecognized assembly machinery for the 4(3H)-quinazolinone scaffold exists.

The following information has been garnered from previous gene knockout (KO) and transcriptional analysis results from the chrysogine producers *Penicillium chrysogenum* and *Fusarium graminearum*:[18,19] (1) a two-module NRPS (ChyA, $A_1$-$T_1$-$C_1$-$A_2$-$T_2$-$C_T$)-

containing gene cluster (*chy* cluster, Fig. 2a) has been confirmed to be responsible for the synthesis of **1**; (2) the chemically synthesized compound L-alanyl-anthranilamide **2** has been demonstrated to be the on-pathway intermediate of **1** via chemical complementation in the *chyA* KO mutant (Fig. 2b); (3) individual deletion of genes *chyC*, *chyE*, and *chyH* does not completely abolish the production of **1**; however, it leads to the production of many shunt products (Supplementary Table 2), which indicates a complex production network of **1** in *P. chrysogenum*; and (4) the in vivo overexpression (OE) of the NRPS gene *chyA* in the *chy* cluster deletion strain of *P. chrysogenum* leads to the accumulation of L-alanyl-anthranilate

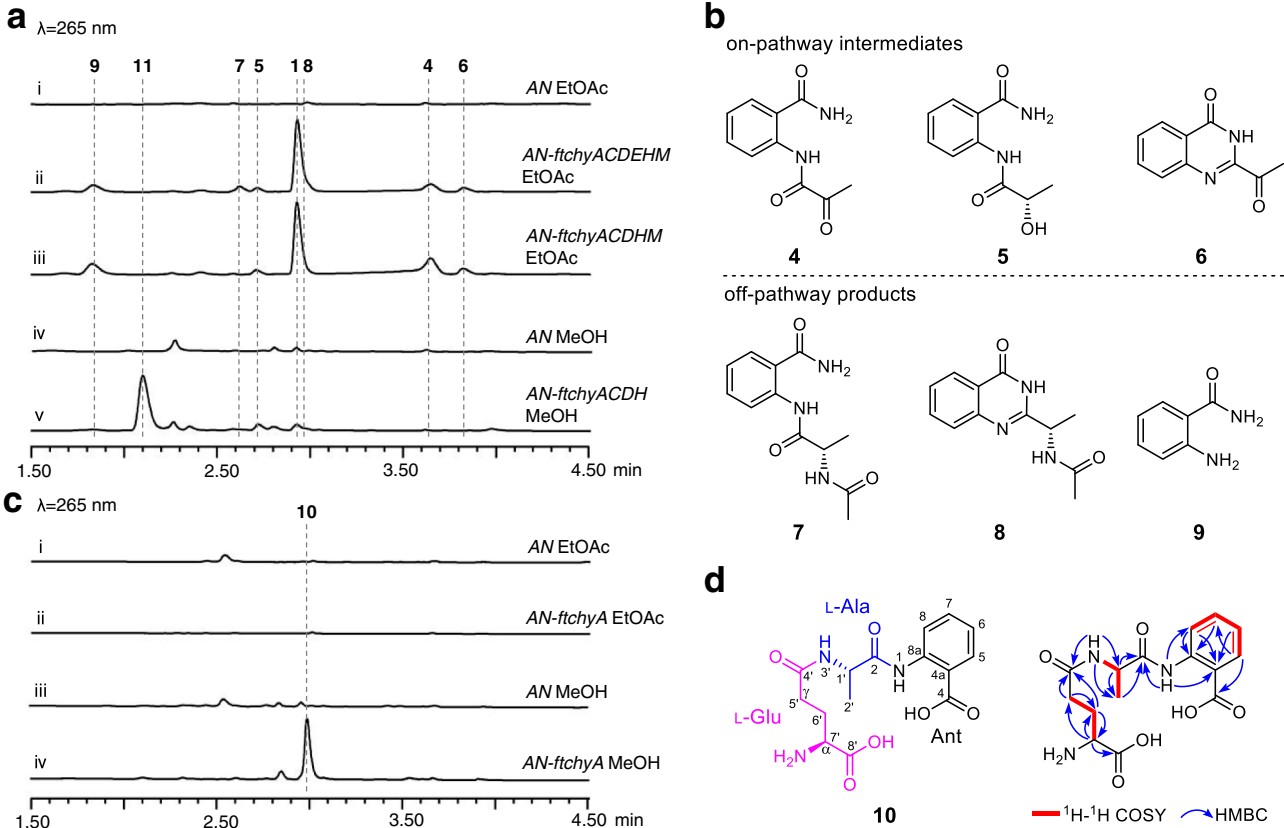

**Fig. 3 | Confirmation of the *ftchy* cluster and heterologously produced products. a** LC-MS analyses of the *A. nidulans* transformant culture extracts. **b** Chemical structures of the compounds isolated from the *AN-ftchyACDEHM*

transformant. **c** LC-MS analyses of the *AN-ftchyA* transformant culture extracts. **d** Chemical structure of **10** showing it is a linear γ-L-glutamyl-L-alanyl-anthranilate tripeptide.

dipeptide **3** and its malonyl- and glutaminyl-tailoring products (Fig. 2b). Although the abovementioned results confirm the role of the *chy* cluster and preliminarily propose the biosynthetic nodes of **1** (Fig. 2b), two important issues remain unsolved: (1) the mechanism by which NRPS ChyA synthesizes **3** and its two modified products; and (2) the chemical conversion that incorporates an amino group into **2** and the subsequent cyclization steps to generate 4(3H)-quinazolinone.

In this work, via in vitro investigation of the pathway of **1**, we discover and demonstrate unexpected assembly machinery for the synthesis of the 4(3H)-quinazolinone scaffold, which mainly includes (1) a two-module NRPS ftChyA, which unexpectedly synthesizes a linear γ-L-glutamyl-L-alanyl-anthranilate tripeptide **10** featuring a rare amide bond constituted by the γ-COOH of L-glutamic acid (L-Glu) and the α-NH2 of L-alanine (L-Ala); (2) an unusual NRPS $C_T$ domain, which catalyses release of the online T2 domain-tethered γ-L-glutamyl-L-alanyl dipeptide via the offline Ant; (3) a class II amidotransferase ftChyD, which uses inorganic ammonium ions or the amide of L-glutamine (L-Gln) to catalyse the amidation of **10** to yield **11**, demonstrating the N-3 nitrogen source for 4(3H)-quinazolinone; and (4) an unusual α-KGD ftChyM, which catalyses the C-N bond oxidative cleavage of **11** to form **4**, where the resultant α-carbonyl group in **4** greatly promotes final spontaneous cyclization to form the 4(3H)-quinazolinone scaffold **6**.

## Results and discussion

### Genome mining reveals the *ftchy* cluster in *F. tricinctum*

To clarify the assembly machinery for the 4(3H)-quinazolinone scaffold synthesis of **1**, we mined the *chy* homologue gene cluster from the *Fusarium* sp. genome database in our lab. A highly homologous cluster (*ftchy*) was identified from *F. tricinctum* CGMCC 3.4731 (Fig. 2a and Supplementary Table 3). The *ftchy* cluster shares similar gene

composition and organization with the *chy* cluster from *P. chrysogenum* and *F. graminearum*. To confirm that the *ftchy* cluster is responsible for the synthesis of **1**, we constructed gene expression combination plasmids for heterologous production in *A. nidulans* (Supplementary Fig. 4). As shown in Fig. 3a, i, ii, **1** was successfully produced and purified from the *AN-ftchyACDEHM* transformant, and its structure was confirmed by NMR analyses (Supplementary Table 5 and Supplementary Figs. 46, 47).

In addition to **1**, six more compounds (**4–9**) were produced by strain *AN-ftchyACDEHM* (Fig. 3a, ii), albeit with lower yields. We purified these compounds from the large-scale fermentation of *AN-ftchyACDEHM* and confirmed their structures by NMR analyses (Supplementary Tables 8–13 and Supplementary Figs. 58–69), which are shown in Fig. 3b. Feeding the *ftchyA* KO strain (*AN-ftchyCDEHM*) with these compounds demonstrated that **4**, **5** and **6** could restore the production of **1**, whereas **7**, **8** and **9** were all off-pathway products (Supplementary Fig. 5). Although the previously identified on-pathway intermediate **2** was not detected in strain *AN-ftchyACDEHM*, the production of **7** strongly indicated that **2** should be converted to **7** by an unknown acetyltransferase of *A. nidulans*[20].

Structural analysis indicated that **1** and **5** are the reduced products of **6** and **4**, respectively, the reaction of which is possibly catalysed by the short chain dehydrogenase/reductase (SDR) encoded by the gene *ftchyC*. Indeed, when purified ftChyC (~31 kDa, from *Escherichia coli*, Supplementary Fig. 6) was incubated with **6** and **4** in the presence of the cofactor NADPH, **1** and **5** were detected (Supplementary Fig. 7). However, reverse dehydrogenation by ftChyC towards **1** and **5** was not observed (Supplementary Fig. 7). These results show that ftChyC is a promiscuous reductase that can reduce **6** and **4**, and the reduction of **6** to **1** is possibly the final step in the synthesis of **1**.

## In vivo heterologous expression of *ftchyA* in *A. nidulans* unexpectedly produces linear tripeptide 10

To dissect the synthetic steps of **1**, we first focused on the two-module NRPS ftChyA. The gene *ftchyA* was cloned from the gDNA of *F. tricinctum* and heterologously expressed in *A. nidulans* under the control of the *gpdA* promoter (Supplementary Fig. 4). After 3.5 days of solid medium culture followed by extraction with ethyl acetate (EtOAc), however, no expected product, such as **3**, was detected (Fig. 3c, i, ii). Alternatively, we used methanol (MeOH) as the solvent to extract the culture, and one compound (**10**) with *m/z* 338 [M + H]$^+$ was produced by *AN-ftchyA* (Fig. 3c, iii, iv). Purification of **10** from the large-scale fermentation of *AN-ftchyA* and subsequent structural determination by NMR analyses (Supplementary Table 14 and Supplementary Figs. 70–75) confirmed that **10** was a linear tripeptide consisting of L-Glu, L-Ala and Ant (Fig. 3d).

It is worth mentioning that, according to the key heteronuclear multiple bond correlation (HMBC) of NH-3′ with C-1′, C-4′ and C-2′ (Fig. 3d), **10** features a rare amide bond constituted by the γ-COOH of L-Glu and the α-NH$_2$ of L-Ala. This type of amide bond is seldom found in nonribosomal peptides, and sporadic examples such as butirosin and microcystin LR from bacteria and δ-L-α-aminoadipyl-L-cysteinyl-D-valine (ACV tripeptide, a penicillin precursor) from fungi, have been reported[21–23]. The production of **10** by *AN-ftchyA* shows an unusual example in which fungal two-module NRPS synthesizes a tripeptide in vivo[24], which is significantly different from the classical fungal two-module NRPSs that usually generate diketopiperazine backbones[25–29].

### ftChyA biochemical assays unveil an unusual two-module NRPS assembly process for the synthesis of 10 and 3

To clarify the unusual relationship between ftChyA and **10**, we next attempted to investigate the function of ftChyA in vitro. Intron-free *ftchyA* was cloned, and ftChyA was expressed and purified from *E. coli* (~265 kDa, Supplementary Figs. 6, 8). *Apo*-ftChyA was enzymatically converted into its *holo* form by using phosphopantetheinyl transferases NpgA and CoA[30]. When 5 µM *holo*-ftChyA was incubated with 1 mM of the substrates L-Glu, L-Ala and Ant, as well as the cofactors ATP and Mg$^{2+}$, two products were produced. The major one is consistent with standard **10**; another minor one with *m/z* 209 [M + H]$^+$ is consistent with expected **3** (Fig. 4a, i, ii, vi). Moreover, **10** was stable under the reaction conditions (Supplementary Fig. 9) indicating that the minor compound is not from **10**; it is also the product of ftChyA. Additional combinations showed that (1) the removal of L-Glu only abolished the production of **10** (Fig. 4a, iii); and (2) the elimination of L-Ala or Ant abolished both **10** and the minor compound (Fig. 4a, iv, v). These results clearly demonstrated that L-Ala and Ant are the building blocks of the minor compound. The minor compound was finally purified from large-scale biochemical assays, and its structure was confirmed to be L-alanyl-anthranilate dipeptide **3** by NMR analyses (Supplementary Table 7 and Supplementary Figs. 53–57).

The above results (1) establish the connection of ftChyA with **10** and represent the biochemical evidence that fungal two-module NRPS synthesizes a tripeptide; and (2) demonstrate that ftChyA has an additional ability to form dipeptide **3**; however, under the equivalent supply of all substrates, ftChyA favours the synthesis of tripeptide **10**. This might be the reason why **3** was not detected in *AN-ftchyA* in vivo. Therefore, we additionally fed Ant (200 µM to 1 mM) into *AN-ftchyA*, and **3** was detected by LC-MS analysis (Supplementary Fig. 10).

### ftChyA-C$_T$ catalyses the release of the online γ-L-glutamyl-L-alanyl-*S*-T$_2$ or L-alanyl-*S*-T$_2$ via the offline anthranilate

Simultaneous production of **10** and **3** by ftChyA implies its unique catalytic mechanism, especially amide bond formation in **10** and **3**, for linear tripeptide and dipeptide synthesis. Phylogenetic analysis (Fig. 4b) and sequence similarity network (SSN) analysis

(Supplementary Fig. 11) of ftChyA-C$_1$ and ftChyA-C$_T$ with other identified fungal NRPS C domains[11,25,31–36] showed that (1) ftChyA-C$_1$ contains the standard H$_{986}$H$_{987}$xxxD$_{991}$ motif (Supplementary Fig. 12), which belongs to the canonical extension clade of amide bond formation; and (2) ftChyA-C$_T$ and its homologous enzymes from other *chy* clusters (Supplementary Fig. 13) feature the P$_{2074}$H$_{2075}$xxxD$_{2079}$ motif (where the first residue His was replaced by Pro; Supplementary Fig. 14) and are clustered into an independent clade with an unknown function.

To clarify the roles of these two C domains, we carried out site-directed mutagenesis on the active residues H$_{987}$ of ftChyA-C$_1$ and H$_{2075}$ of ftChyA-C$_T$ (Supplementary Figs. 6, 8). The in vitro assays showed that (1) the mutation H$_{987}$A of ftChyA-C$_1$ abolished the formation of **10**; however, it did not affect the production of **3** (Fig. 4c, i); and (2) ftChyA-C$_T$ H$_{2075}$A abolished the production of **10** and **3** (Fig. 4c, ii). To exclude the possible complementary effects of H$_{986}$ on the H$_{987}$A mutation in ftChyA-C$_1$, we further constructed the A$_{986}$A$_{987}$xxxA$_{991}$ (ftChyA-C$_1$*) mutant; however, ftChyA-C$_1$* retained the ability to generate **3** (Fig. 4c, iii). These results (1) demonstrate that ftChyA-C$_1$ does not participate in the synthesis of **3**, however, it is responsible for the generation of **10**. Thus, ftChyA-C$_1$ is responsible for amide bond formation between L-Glu and L-Ala of **10**; (2) strongly indicate that ftChyA-C$_T$ not only catalyses amide bond formation between L-Ala and Ant of **10** and **3** but also catalyses the release of **10** and **3**.

Based on the canonical NRPS assembly rule and the order of amino acids in **10**, ftChyA-A$_1$ should recognize L-Glu for the synthesis of **10** (Fig. 5a). However, for the synthesis of **3**, ftChyA-A$_1$ needs to recognize L-Ala (Fig. 5a). Therefore, the function of ftChyA-A$_1$ for **10** and **3** seems inconsistent. Moreover, the chain transfer from γ-L-glutamyl-L-alanyl-*S*-T$_2$ to γ-L-glutamyl-L-alanyl-*S*-T$_1$, as well as the repeat recognition of Ant by ftChyA-A$_2$, are both required to support the synthesis of **10** (Fig. 5a). Therefore, the proposed mechanism a for **10** and **3** is questionable. Alternatively, according to the recently reported unusual pass-back mechanism of NRPS from α-proteobacteria[37], mechanism b (no chain transfer involved) was also proposed (Fig. 5b). In this model, ftChyA-A$_2$ uniformly recognizes L-Ala for **10** and **3**; however, ftChyA-A$_1$ needs to recognize L-Glu and Ant for **10** and Ant for **3**. It is worth mentioning that although mechanisms a and b are different, the releases of the proposed γ-L-glutamyl-L-alanyl-anthranilyl-*S*-T and L-alanyl-anthranilyl-*S*-T to forms **10** and **3** are all mediated by water.

To test these two hypotheses, we first expressed and purified stand-alone ftChyA-A$_1$ (~86 kDa, Supplementary Fig. 6) from *E. coli*, which was then subjected to an ATP-inorganic pyrophosphate(PPi) release assay[38]. In contrast to L-Ala and Ant, ftChyA-A$_1$ favoured L-Glu, where the specificity difference was significant (Fig. 4d). Additional matrix-assisted laser desorption/ionization-time of flight mass spectrometry (MALDI-TOF MS) analysis showed that, under the equivalent competitive experiment, ftChyA-A$_1$ only recognized and loaded L-Glu to the Ser binding site (S$_{800}$) of ftChyA-T$_1$-*holo* (Fig. 4d). The subsequent ATP-PPi release assay of ftChyA-A$_2$ (~108 kDa, with a maltose binding protein (MBP)-tag, Supplementary Fig. 6) towards L-Ala and Ant showed that L-Ala was the substrate of ftChyA-A$_2$ (Fig. 4d). These results demonstrate that both mechanisms a and b are not reasonable for **10** and **3**, thus excluding that the formation of **10** and **3** was mediated by water. Indeed, when the biochemical assays of ftChyA were carried out in H$_2$$^{18}$O-Tris-HCl buffer, the incorporation of $^{18}$O into **10** and **3** was not observed, and the molecular weights of **10** and **3** were not increased (Fig. 4e).

Considering the results of C domain and A domain, a model for the synthesis of **10** and **3** by ftChyA was proposed (Fig. 5c). In this model, Ant does not need to be loaded by ftChyA-A$_1$ or ftChyA-A$_2$ to the T domains; the online γ-L-glutamyl-L-alanyl-*S*-T$_2$ or L-alanyl-*S*-T$_2$ is attacked by the offline Ant to release **10** or **3**, respectively, where the process is catalysed by ftChyA-C$_T$. To test this hypothesis, blocking the carboxyl group of Ant is needed, which rules out the possibility of A domains loading the substrate. The ideal compound

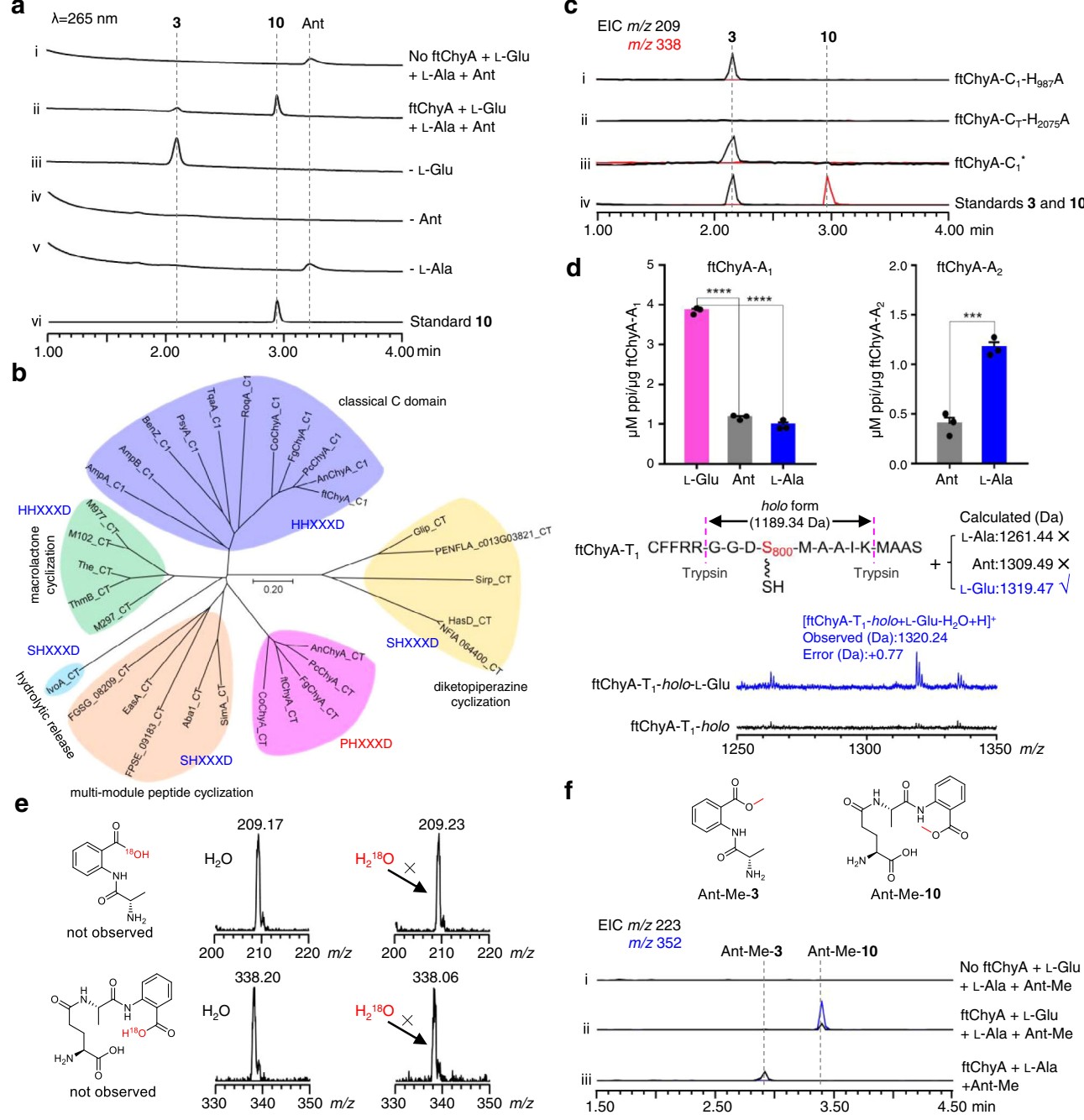

**Fig. 4 | Biochemical confirmation of the unexpected assembly process of ftChyA for the synthesis of 10 and 3. a** Biochemical confirmation of ftChyA synthesizing **10** and **3**. **b** Phylogenetic analysis of fungal NRPS C domains showing that ftChyA-C_T is separated into an independent clade. The C domain protein sequences used for phylogenetic analysis are listed in Source Data file. **c** ftChyA-C_1 and ftChyA-C_T domain mutations confirm that the C_1 domain is responsible for amide bond formation between L-Glu and L-Ala of **10** and that C_T is responsible for amide bond formation between L-Ala and Ant to release **10** and **3**. **d** ATP-PPi release assay and

MALDI-TOF MS analysis of the A domains for substrate recognition. Data are shown as the mean ± SEM of 3 independent experiments. ***$P < 0.001$, ****$P < 0.0001$ (ftChyA-A_1, $P = 1.5e$-6 between L-Glu and Ant, $P = 4.5e$-6 between L-Glu and L-Ala; ftChyA-A_2, $P = 0.0008$ between L-Ala and Ant), unpaired two-tailed Student's $t$ test. **e** Biochemical assays of ftChyA in $H_2^{18}O$-Tris buffer confirming that water is not involved in the formation of **10** and **3**. **f** In vitro assays of ftChyA with L-Glu, L-Ala and Ant-Me. The extracted ion chromatograms (EICs) were extracted at $m/z$ 223 [M + H]$^+$ for Ant-Me-**3** and $m/z$ 352 [M + H]$^+$ for Ant-Me-**10**.

is anthranilate methyl ester (Ant-Me). When Ant was replaced by Ant-Me in ftChyA assays, the corresponding products Ant-Me-**10** and Ant-Me-**3** were detected by LC-MS and HRMS analyses (Fig. 4f, Fig. 5c and Supplementary Figs. 44, 45), respectively. We further constructed, expressed, and purified C_T domain-truncated ftChyAΔC_T (A_1-T_1-C_1-A_2-T_2, ~218 kDa, Supplementary Figs. 6, 8) and stand-alone ftChyA-C_T (~55 kDa, Supplementary Fig. 6) from *E. coli*. When ftChyAΔC_T was incubated with ftChyA-C_T, the formation of **10**

and **3** was observed (Supplementary Fig. 15). The yields of **10** and **3** increased when the concentration of ftChyA-C_T increased (Supplementary Fig. 15). These results demonstrate that ftChyA-C_T is a unique C domain, representing an unusual function of fungal two-module NRPS C domains[34]. The attack by the offline free amino acid on the online T domain-tethered dipeptide or T domain-tethered amino acid to release the formation of the linear tripeptide or dipeptide is catalysed.

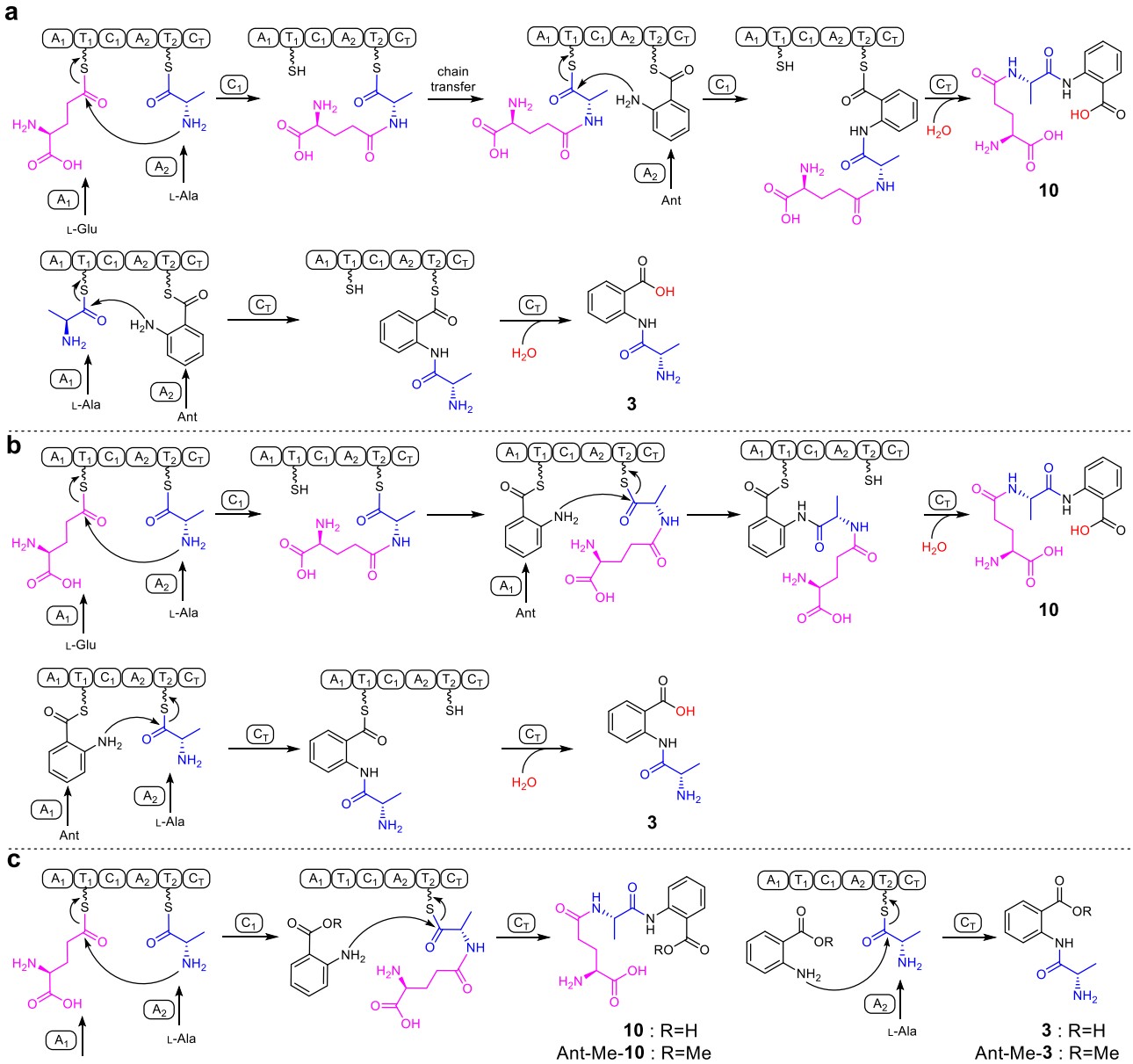

**Fig. 5 | Three assembly mechanisms of NRPS ftChyA were proposed for the synthesis of 10 and 3. a** Canonical assembly rule of NRPS. **b** Pass-back mechanism of NRPS. **c** Proposed mechanism in this study.

## ftChyD uses inorganic ammonium ions or the amide of L-Gln to catalyse the amidation of 10 to 11 and 3 to 2

Confirmation of ftChyA as an unusual tripeptide synthase encouraged us to investigate the next steps of **1**. The next focus is the amidation of the Ant fragments of **10** and **3**, where the candidate enzyme for this modification is ftChyD. ftChyD is an asparagine synthase protein belonging to the class II glutamine amidotransferase family;[39] it contains an intact N-terminal catalytic cysteine (Cys$_2$) residue and C-terminal synthase domain to bind ATP and substrate (Supplementary Fig. 16)[40].

To investigate the ability of ftChyD (~78 kDa) to catalyse the amidation of **10** and **3**, it was expressed and purified from *E. coli* (Supplementary Fig. 6). With L-Gln and ATP, ftChyD converted **10** and **3** into the corresponding amidation products **11** ($m/z$ 337 [M + H]$^+$) and **2** ($m/z$ 208 [M + H]$^+$), respectively (Fig. 6a, i, ii, and Supplementary Fig. 17a). These two products were purified from the large-scale in vitro assays, and their structures were confirmed by

NMR analyses (Supplementary Tables 6, 15 and Supplementary Figs. 48–52 and Figs. 76–81). Further $^{15}$N-labelled L-Gln assays showed that the amide of L-Gln was the amide donor (Fig. 6b, i, ii, and Supplementary Fig. 17b–d), whereas the α-NH$_2$ of L-Gln could not be incorporated into **11** and **2** (Fig. 6b, iii, and Supplementary Fig. 17b–d). Interestingly, when we used NH$_4$Cl to replace L-Gln, the formation of **11** and **2** was enhanced (Fig. 6a, iii, and Supplementary Fig. 17a), supporting that ftChyD favours inorganic ammonium ions as the amide donor. The incorporation of NH$_4^+$ into **11** and **2** was also demonstrated by $^{15}$NH$_4$Cl (Fig. 6b, iv, and Supplementary Fig. 17b–d). The $kcat/K_M$ calculation showed that the catalytic efficiency of ftChyD towards **10** was nearly 36-fold greater than that towards **3** (Supplementary Fig. 18), which indicates that **10** is the optimal substrate of ftChyD. These results demonstrate that the N-3 of the 4(3H)-quinazolinone scaffold in **1** comes from the inorganic ammonium ions or the amide-N of L-Gln, which is different from the mechanisms shown in Fig. 1c, d.

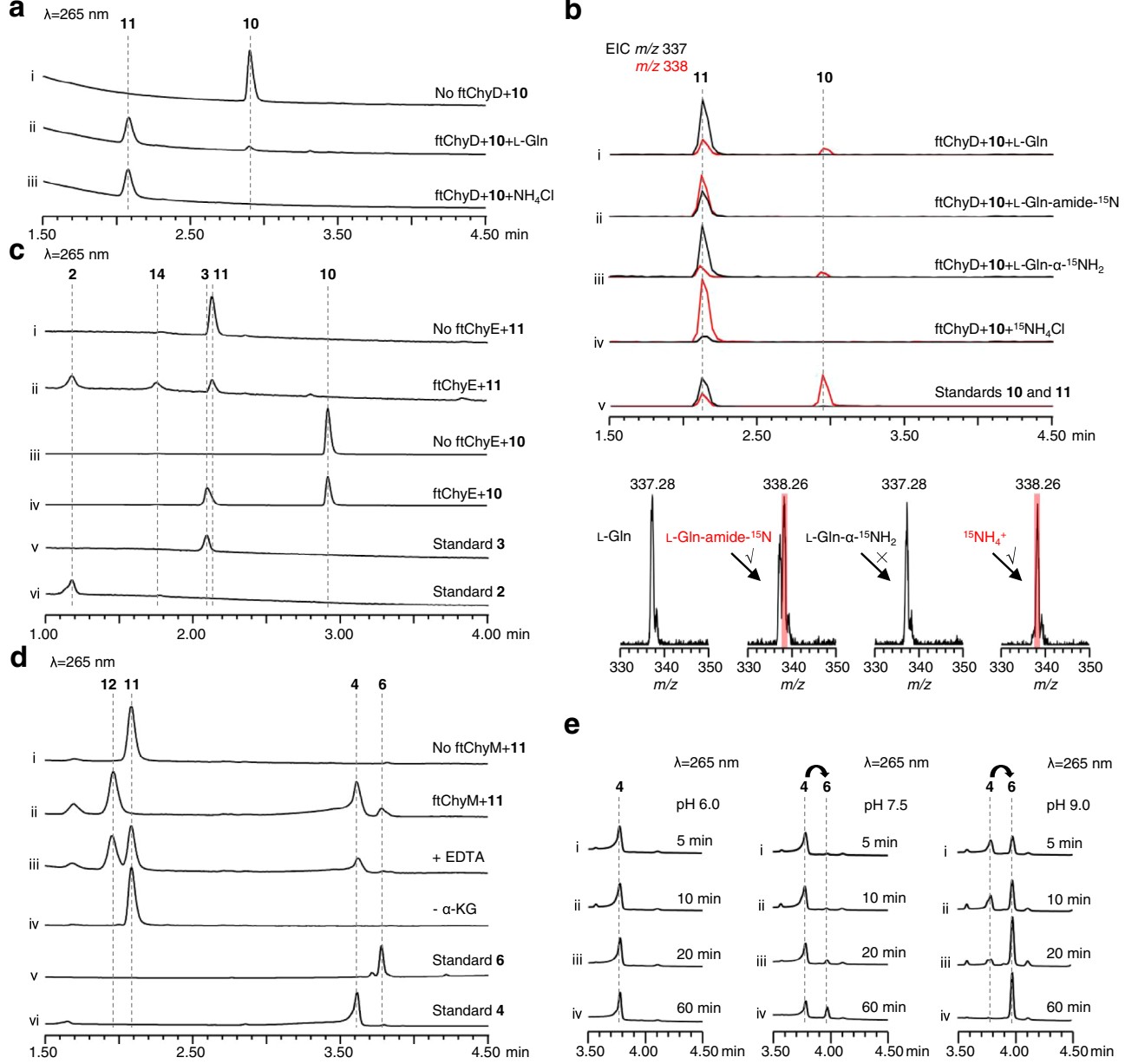

**Fig. 6 | Biochemical confirmation of the functions of ftChyD, ftChyE and ftChyM. a** ftChyD uses $NH_4Cl$ or ʟ-Gln to catalyse the amidation of **10** to form **11**. **b** LC-MS analysis of the incorporation of $^{15}N$ into **11**. The EICs were extracted at $m/z$ 338 [M + H]⁺ for **10** and $^{15}N$-labelled **11**, and $m/z$ 337 [M + H]⁺ for **11**. **c** ftChyE catalyses the hydrolysis of **11** and **10** to form **2** and **3**, respectively. **d** ftChyM is an α-KGD that catalyses the C-N bond oxidative cleavage of **11** to form **4** via a possible intermediate **12**. **e** Conversion of **4** to **6** shows the alkaline-induced spontaneous C-2-N-3 bond closure.

## ftChyE is a tripeptide hydrolase that converts 11 to 2

A previous feeding experiment showed that **2** is the precursor of **1**[18]; thus, removal of the ʟ-Glu fragment of **11** by hydrolysis to give **2** should be investigated. We focused on ftChyE, although it was previously annotated as a malonyl transferase[18]. However, conserved domain analysis showed that it is the glutaminase subunit of a class I glutamine amidotransferase (GATase)[39]. Sequence alignment of ftChyE with its homologous enzymes and other identified class I GATases showed that ftChyE contains the intact catalytic triad ($Cys_{102}$-$His_{189}$-$Glu_{191}$, C-H-E, Supplementary Fig. 19) for glutaminase activity, which hydrolyses glutamine to glutamate and nascent ammonia[41].

To test whether ftChyE could hydrolyse **11** to **2**, ftChyE (~74 kDa, with an MBP tag, Supplementary Fig. 6) was expressed and purified from *E. coli*. When **11** was incubated with ftChyE, the production of **2** (and its spontaneous cyclization product **14**, see below) was detected (Fig. 6c, i, ii). The substrate tolerance of ftChyE

hydrolysing **10** to give **3** was also investigated (Fig. 6c, iii, iv). Confirmation of the function of ftChyE (1) establishes the bridge between the tripeptide and dipeptide in the synthesis of **1** and (2) represents the unique function of class I GATase as tripeptide hydrolase.

## ftChyM is an unusual α-KGD that catalyses the C-N bond oxidative cleavage of 11 to form 4

Although ftChyE-catalysed hydrolysis of **11** to **2** was confirmed, previous transcription analysis of the *chy* cluster in *F. graminearum* showed that *chyE* was expressed at a lower level than the remaining cluster genes[19], suggesting that *chyE* is not the essential gene for the synthesis of **1**. To test this hypothesis, we constructed the *AN-ftchyACDHM* transformant and found that elimination of *ftchyE* does not abolish the production of **1** (Fig. 3a, iii), which suggests that the ftChyE-participating pathway (via **2**, including **11 → 2** and **10 → 3 → 2**) is

**Fig. 7 | The proposed complex pathways for generating the 4(3H)-quinazolinone scaffold in 1 synthesis.** The primary pathway shows unexpected assembly machinery starting from a linear tripeptide and the salvage pathway depends on the promiscuous substrate selectivity of *ftchy* cluster enzymes.

possibly not the primary route but the salvage route for the synthesis of **1** (Fig. 7).

Since **4** is the on-pathway intermediate of **1** (Fig. 3b and Supplementary Fig. 5), a C-N bond cleavage process from **11** to **4** should occur (Fig. 7). Through this pathway, the oxygen atom of the α-OH group (C-1′-OH) of **1** could be from water (Fig. 7 and Supplementary Fig. 25a). To test this hypothesis, we incubated *AN-ftchyACDEHM* in $H_2^{18}O$-medium, and the incorporation of $^{18}O$ into **1** was observed by LC-MS (Supplementary Fig. 20). The remaining unidentified genes in the *ftchy* cluster were *ftchyM* and *ftchyH*. Conserved domain analysis showed that ftChyH is a flavin-dependent oxidase that contains a berberine bridge enzyme (BBE) conserved domain (Supplementary Fig. 21a) and belongs to the BBE-like oxidase superfamily[42]. BBE-like oxidases usually catalyse dehydrogenation-mediated C-C or C-N bond formation reactions during natural product biosynthesis[42]. Therefore, ftChyH was proposed to be responsible for 4(3H)-quinazolinone ring formation via C-2-N-3 bond closure. On the other hand, ftChyM showed 31%/45% identity/similarity to GA₄ desaturase, an α-KGD that converts $GA_4$ to $GA_7$ by forming a C-1-C-2 double bond during gibberellic acid biosynthesis[43]. Conserved domain analysis showed that the HxD motif (which binds $Fe^{2+}$) and the RxS motif (which binds α-ketoglutarate, α-KG) were intact in ftChyM (Supplementary Fig. 21b). Further sequence clustering analysis showed that ftChyM and its homologous family proteins represent a unique α-KGD cluster in fungi (Supplementary Fig. 22). Therefore, we reasoned that ftChyM might catalyse the C-N bond oxidative cleavage of **11** to form **4**.

To test this hypothesis, ftChyM (~46 kDa, Supplementary Fig. 6) was expressed in *E. coli* and purified by using dithiothreitol (DTT; 1 mM final concentration)-containing Tris-HCl buffer, which ensured that the Fe$^{II}$ core was not inactivated due to oxidation. When ftChyM was

incubated with **11** in the presence of α-KG, $Fe^{2+}$ and vitamin C, **4** and **6** were produced (Fig. 6d, i, ii). The addition of EDTA (5 mM) to bind $Fe^{2+}$ greatly decreased the activity of ftChyM (Fig. 6d, iii). Moreover, the elimination of α-KG and formation of **4** and **6** were not observed (Fig. 6d, iv), thus suggesting that ftChyM, as a standard α-KGD, requires these cofactors for enzymatic activity. We also investigated the ftChyM-catalysed deamination of **2**, and the time-course analysis showed that the chemical conversion of ftChyM towards **2** is slower than that towards **11** (Supplementary Fig. 23a, c). Interestingly, when **5** was incubated with ftChyM, albeit with low yields, the formation of **4** and **6** was also observed (Supplementary Fig. 23b), which showed that ftChyM has an additional ability to catalyse dehydrogenation. These results demonstrate that ftChyM-catalysed conversion of **11** to **4** is the indispensable route for **1** synthesis. Indeed, further elimination of the *ftchyM* transformant *AN-ftchyACDH* led to the accumulation of **11** but abolished the production of **1** (Fig. 3a, iv, v).

Apart from **4** and **6**, compound **12** with *m/z* 353 [M + H]⁺ (**11** plus 16 Da) was observed in the ftChyM-catalysed assay (Fig. 6d). We initially attempted to purify **12**; however, it spontaneously and completely converted to **4** and **6** within 8 h (Supplementary Fig. 23c, d). When the ftChyM-catalysed assay was carried out in ~80% $H_2^{18}O$-Tris HCl buffer, nearly ~73% of $^{18}O$-labelled **4** and **6** (in sum) was detected (Supplementary Fig. 24); however, the incorporation of $^{18}O$ into **12** was not observed (Supplementary Fig. 24). Therefore, **12** should be the hydroxylation intermediate of **11** catalysed by ftChyM, which is possibly formed by the following steps (Fig. 7 and Supplementary Fig. 25a, b): (1) the reactive [Fe$^{IV}$ = O] species of ftChyM homolytically breaks the inactive Cα-Hα bond of the L-Ala moiety in **11** to produce the **11** Cα radical and (2) the reduced Fe$^{III}$-OH provides a hydroxyl radical to yield hemiaminal intermediate **12**[44,45]. Once **12** is produced, its dehydration

to form iminium intermediate **13**, as well as the subsequent hydrolysis of **13** to give **4**, are likely spontaneous steps (Fig. 7 and Supplementary Fig. 25a, b)[44,46].

### The α-carbonyl group in 4 greatly promotes spontaneous cyclization to form the 4(3*H*)-quinazolinone scaffold

Although **4** and **6** were simultaneously produced in the ftChyM-catalysed assays, we reasoned that **6** is the spontaneous cyclization product of **4**. This is possibly due to the electron-withdrawing effect of the α-carbonyl group of **4**, which makes the C-2 centre more electron deficient, thereby inducing C-2-N-3 closure to form the 4(3*H*)-quinazolinone scaffold (Supplementary Fig. 25c). To test this hypothesis, **4** was incubated alone in Tris-HCl buffer at a pH of 7.5, and the time-dependent formation of **6** was clearly observed (Supplementary Fig. 26a). The addition of ftChyM did not accelerate this conversion (Supplementary Fig. 26b), which confirms that ftChyM does not participate in the subsequent cyclization step (Fig. 7). Notably, the conversion rate of **4** to **6** occurs with an increase in the pH value of the buffer (Fig. 6e), which indicates that the hydroxyl anion from the alkaline buffer might be beneficial for the proposed C-2-OH departure (Supplementary Fig. 25c).

We also investigated the spontaneous transformations of **2** and **5** into their corresponding cyclization products **14** and **1**, respectively (Fig. 7, Supplementary Table 16 and Supplementary Figs. 82, 83). Although these cyclization products were observed, the conversions were slow (Supplementary Fig. 27), which shows that the α-carbonyl group of **4** greatly promotes spontaneous cyclization for the synthesis of 4(3*H*)-quinazolinone scaffold **6**.

### ftChyH ensures the supply of 4 by correcting the additional reduction reaction performed by ftChyC

The above results confirmed that the formation of the 4(3*H*)-quinazolinone scaffold in **1** is an α-carbonyl group formation-driven nonenzymatic cyclization process. This observation excludes the BBE oxidase ftChyH, the ideal candidate that was previously proposed to be responsible for this key step. Thus, the actual function of ftChyH should be reconsidered. We constructed *AN-ftchyACDM* and found that **5** accumulated in this transformant (Supplementary Fig. 28a), which highly suggests that ftChyH catalyses the dehydrogenation of **5** to give **4**.

To test this hypothesis, we attempted to confirm the function of ftChyH using a purified enzyme; however, this protein was not soluble in *E. coli* even when glutathione *S*-transferase (GST)-tagged or MBP-tagged ftChyH was constructed (Supplementary Fig. 29). Alternatively, yeast was used as the heterologous expression host, and the ftChyH cell-free extracts could catalyse the dehydrogenation of **5** to form **4** and **6** (Supplementary Fig. 28b). When **5** was replaced by **11** or **2**, the generation of **4** and **6** was not observed (Supplementary Fig. 28b). These results confirm that ftChyH only catalyses the dehydrogenation reaction and can correct the additional reduction of ftChyC towards **4**, ensuring the primary pathway (**4** → **6**) in the quick construction of the 4(3*H*)-quinazolinone scaffold. Other branched pathways, depending on the nonenzymatic cyclization (**5** → **1**, Supplementary Fig. 27) or promiscuous substrate selectivity of ftChyM (**14** → **6** → **1**, Supplementary Fig. 30), were also confirmed during the synthesis of **1** (Fig. 7).

As shown in Fig. 7, synthesis of **1** from **10** represents an alternative route for generating the 4(3*H*)-quinazolinone scaffold. Apart from the inorganic substrates are involved in **1** synthesis, the main unexpected assembly machinery is that L-Glu is first recruited by ftChyA to synthesize **10**, however, it is then removed as L-Gln by ftChyM during the post tailoring steps of **1**. This seems a redundant process for the formation of **1**; however, it is particularly worth mentioning that the generated L-Gln from ftChyM reaction could be recaptured and hydrolysed by ftChyD to yield L-Glu and ammonium ions, where these two products could re-participate in ftChyA-catalysed **10** formation

and ftChyD-catalysed amidation reactions, respectively. Therefore, from this perspective, an efficient self-circulation system among ftChyA, ftChyD and ftChyM-catalysed reactions has been established during the synthetic process of **1** (Supplementary Fig. 84).

In this work, via in vitro investigation of the pathway of **1**, unexpected assembly machinery for the synthesis of the 4(3*H*)-quinazolinone scaffold was unveiled and biochemically confirmed, which importantly (1) reveals a fungal two-module NRPS with an unusual $C_T$ domain catalysing tripeptide formation; (2) reveals that the nitrogen source for N-3 is inorganic ammonium ions or amide of L-Gln; and (3) demonstrates an unusual α-KGD catalysing the C-N bond oxidative cleavage of a tripeptide to form a dipeptide. Our study demonstrates a unique release and tailoring mechanism of nonribosomal peptides and represents an alternative native synthetic logic in the construction of 4(3*H*)-quinazolinone scaffolds.

## Methods

### General methods

Reagents were purchased from Thermo Fisher Scientific, Sigma-Aldrich or New England BioLabs. Primer synthesis and DNA sequencing were performed by the Sangon Biotech Co., Ltd. (Shanghai, China). The primer sequences and plasmids used in this study are provided in the Source Data file and Supplementary Table 1, respectively. LC-MS analyses were performed on a Waters ACQUITY H-Class UPLC-MS system coupled to a PDA detector and an SQD2 MS detector with an ESI source. Chromatographic separation was performed at 35 °C using a C18 column (ACQUITY UPLC® BEH, 1.7 μm, 2.1 × 100 mm, Waters). LC-MS analyses were performed on a Waters UPLC-MS system with a linear gradient of 5–99% MeCN-H$_2$O (both with 0.02% v/v formic acid) in 10 min followed by 99% MeCN for 3 min and then 5% MeCN-H$_2$O for 3 min, with a flow rate of 0.4 mL/min. The MS data were collected in the *m/z* range of 150–1500 in positive and negative modes simultaneously. MPLC was performed on a BUCHI Reveleris® X2 Flash Chromatography System with UV and ELSD detectors using a BUCHI Reveleris® C18 column (40 μm, 80 g). Semipreparative HPLC separations were performed on a Shimadzu Prominence HPLC system using a YMC-Pack ODS-A column (5 μm, 10 × 250 mm). NMR spectra were recorded on a Bruker AVANCE III NMR (400 MHz) with a 5 mm broadband probe and TMS as an internal standard. The NMR data analyses of structures were performed using MestReNova12.0.1 software. HRMS data were obtained on a Fourier transform ion cyclotron resonance-mass spectrometer (FT-ICR-MS) (Bruker SolariII, Bremen, Germany) or quadrupole time-of-flight (QTOF) mass spectrometer (Bruker IMPACT II, Bremen, Germany).

### Strains

*Fusarium tricinctum* CGMCC 3.4731 was obtained from the China General Microbiological Culture Collection Centre (CGMCC). *Saccharomyce cerevisiae* BJ5464-NpgA was used as the host for the expression of ftChyH or for the construction of the *A. nidulans* overexpression plasmids through homologous recombination. *Aspergillus nidulans* was used as the host for the heterologous expression of the *ftchy* cluster. *E. coli* XL-1 was used for cloning. *E. coli* BL21 (DE3) was used for protein expression of NpgA, ftChyA and its mutants, ftChyD, ftChyE, ftChyC, ftChyM, ftChyA-A$_1$, ftChyA-A$_2$, ftChyA-A$_1$T$_1$, ftChyAΔC$_T$ and ftChyA-C$_T$.

### Heterologous expression of the *ftchy* cluster in *A. nidulans*

To obtain strains of heterologous expression in *A. nidulans*, 2 μL plasmids (pIM 3201–3206) was added to 100 μL of protoplasts of *A. nidulans*. After incubation on ice for 30 min, 600 μL PEG solution was added to the mixture, which was placed at room temperature for 20 min. The mixture was cultured in regeneration dropout solid CD-SD medium (CD medium with 1.2 mM sorbitol) at 37 °C for 2–3 days. The transformants were moved to solid CD medium at 37 °C for 3–4 days

for sporulation. Then, the spores were incubated in solid CD-ST medium (20 g/L starch, 20 g/L casein hydrolysate (acid), 50 mL/L nitrate salts, 1 mL/L trace elements and 20 g/L agar) at 25 °C for 3.5 days. The products from *AN-ftchyACDEHM*, *AN-ftchyACDHM*, *AN-ftchyACDM* and *AN-ftchyA* were extracted by EtOAc, respectively. The products from *AN-ftchyA* and *AN-ftchyACDH* were extracted by MeOH, respectively. To test whether the oxygen atom of the α-OH group of **1** could be from water, *AN-ftchyACDEHM* was cultured in ~95% $H_2^{18}O$-CD-ST medium and compound **1** was extracted by EtOAc. The organic phase was dried *in vacuo*, and the products were dissolved in methanol for LC-MS analysis.

## Feeding assays of 4–9 in the *AN-ftchyCDEHM* mutant

Recombinant plasmids (pIM 3202-3203) were transformed into *A. nidulans* to obtain the strain *AN-ftchyCDEHM*. The strain was cultured in 4 mL solid CD-ST medium together with 100 μM compound (**4**, **5**, **6**, **7**, **8** or **9**) at 25 °C for 4 days. The products were extracted with ethyl acetate/acetone (v/v, 3/1). The organic phase was dried *in vacuo*, and the products were dissolved in methanol for LC-MS analysis.

## In vitro characterization of ftChyC

To analyse ftChyC-catalysed reduction reactions with substrates **4** and **6**, the in vitro assay was performed in 100 μL buffer C (pH 7.5) containing 4 μM ftChyC, 100 μM substrate (**4** or **6**) and 200 μM NADPH at 25 °C for 3 h. To analyse ftChyC-catalysed dehydrogenation reactions with substrates **1** and **5**, the in vitro assay was performed in 100 μL buffer C containing 4 μM ftChyC, 100 μM substrate (**1** or **5**) and 200 μM NADP at 25 °C for 3 h. The control assays were performed without ftChyC. The reactions were quenched by the addition of 200 μL EtOAc and centrifuged at 17,000 g for 5 min to remove the precipitated proteins. The organic layer was dried *in vacuo*. The residue was dissolved in 200 μL methanol for LC-MS analysis.

## In vitro characterization of ftChyA and its mutants

The purified 5 μM ftChyA was converted to its *holo* form by incubation in 20 mM Tris-HCl, 100 mM NaCl, 20 μM NpgA, 0.1 mM CoA and 10 mM MgCl$_2$·6H$_2$O in a total volume of 50 μL buffer C (pH 7.5) for 1 h at 25 °C. The complete reaction was initiated by the addition of 5 mM ATP, 1 mM L-Glu, 1 mM L-Ala and 1 mM Ant in a final volume of 50 μL at 25 °C overnight. Three reaction mixtures were performed: (1) L-Ala and Ant, (2) L-Glu and L-Ala, and (3) L-Glu and Ant. The control assay was performed without ftChyA. To clarify the roles of these two C domains, in vitro assays of the associated mutants ftChyA-C$_1$-H$_{987}$A (9 μM), ftChyA-C$_T$-H$_{2075}$A (7 μM) and ftChyA-C$_1$* (A$_{986}$A$_{987}$xxxA$_{991}$) (8 μM) for L-Glu, L-Ala and Ant were performed under the same conditions, respectively. To show that the formation of **10** and **3** was not mediated by water, the complete reaction of ftChyA was performed in a total volume of 50 μL ~ 80% $H_2^{18}O$-Tris buffer. To show that Ant does not need to be loaded by ftChyA-A$_1$ or ftChyA-A$_2$ to the T domains, two reaction mixtures were performed: (1) L-Glu, L-Ala and Ant-Me, and (2) L-Ala and Ant-Me. All reactions were quenched by the addition of an equal volume of methanol, and the precipitated protein was removed by centrifugation at 17,000 g for 10 min. The supernatant was subjected to LC-MS or HR-MS analysis.

## In vitro characterization of the ftChyA-C$_T$ domain

The purified 10 μM ftChyAΔC$_T$ was converted to its *holo* form by incubation in ftChyA-C$_T$ (5 μM or 20 μM), 20 mM Tris-HCl (pH 7.5), 100 mM NaCl, 20 μM NpgA, 0.1 mM CoA and 10 mM MgCl$_2$·6H$_2$O in a total volume of 50 μL buffer C (pH 7.5) for 1 h at 25 °C. The reaction was initiated by adding 5 mM ATP, 1 mM L-Glu, 1 mM L-Ala and 1 mM Ant in a final volume of 50 μL at 25 °C for 3 h. The reaction with 1 mM L-Ala and 1 mM Ant was performed under the same conditions. The control assay was performed without ftChyA-C$_T$. The reactions were quenched by adding an equal volume of methanol, and the precipitated protein was

removed by centrifugation at 17,000 g for 10 min. The supernatant was subjected to LC-MS analysis.

## In vitro characterization of ftChyD

The reaction mixture (50 μL) containing 20 μM ftChyD, 100 μM compound **10**, 4 mM ATP, 10 mM MgCl$_2$·6H$_2$O, 1 mM ammonia donor (NH$_4$Cl, $^{15}$NH$_4$Cl, L-Gln, L-Gln-α-$^{15}$NH$_2$ or L-Gln-amide-$^{15}$N) and 20 mM Tris-HCl (pH 7.5) in buffer C (pH 7.5). The mixtures were incubated in 25 °C for 3 h. The control assay was performed without ftChyD. The reaction of ftChyD with **3** was performed at 25 °C overnight. The reactions were quenched by adding an equal volume of methanol, and the precipitated protein was removed by centrifugation at 17,000 g for 10 min. The supernatant was subjected to LC-MS analysis.

## Measurement of the kinetic parameters of ftChyD towards 10 and 3

To determine the kinetic parameters of ftChyD toward **10**, 50 μL reaction mixture containing 2 μM ftChyD, 4 mM ATP, 10 mM MgCl$_2$·6H$_2$O, 1 mM NH$_4$Cl, 20 mM Tris-HCl (pH 7.5) and different concentrations of **10** (0.1, 0.5, 1, 5, 10 and 20 μM) were incubated in 25 °C for 5 min, respectively. To determine the kinetic parameters of ftChyD toward **3**, 50 μL reaction mixture containing 10 μM ftChyD, 4 mM ATP, 10 mM MgCl$_2$·6H$_2$O, 1 mM NH$_4$Cl, 20 mM Tris-HCl (pH 7.5) and different concentrations of **3** (5, 10, 20, 50, 100 and 200 μM) were incubated in 25 °C for 10 min, respectively. 50 μL methanol was added and rigorously mixed by vortexing. After centrifugation at 17000 g for 10 min, 2 μL of liquid supernatant was used for LC-MS analysis and quantified by a standard curve. The kinetics data were fitted to the Michaelis-Menten equation using GraphPad Prism 7 software. For each concentration of substrate, four replicates were performed.

## In vitro characterization of ftChyE

The reaction mixture (50 μL) containing 20 μM ftChyE and 100 μM substrate (**11** or **10**) in buffer C (pH 7.5) and was incubated in 25 °C for overnight. The control assay was performed without ftChyE. The reactions were quenched by adding an equal volume of methanol, and the precipitated protein was removed by centrifugation at 17,000 g for 10 min. The supernatant was subjected to LC-MS analysis.

## In vitro characterization of ftChyM

To show that ftChyM catalyses the C-N bond oxidative cleavage of **11**, the complete reaction mixture (50 μL) for the ftChyM assay containing 5 μM ftChyM, 200 μM compound **11**, 2.5 mM α-ketoglutaric acid, 0.4 mM FeSO$_4$, 4 mM vitamin C and 50 mM Tris-HCl (pH 7.5) in buffer C (pH 7.5) and was incubated in 25 °C for 2 h. To show that ftChyM is a standard α-KGD, two reaction mixtures were performed: (1) + 5 mM EDTA and (2) - α-ketoglutaric acid. The in vitro assays of 20 μM ftChyM with 100 μM substrate (**14** or **5**) were performed under the same conditions. The control assay was performed without ftChyM. To show that ftChyM catalyses the conversion of **11** to form **4** via a spontaneous hydrolysis step, the ftChyM-catalysed assay was performed in ~80% $H_2^{18}O$-Tris-HCl buffer. The reactions were quenched by adding an equal volume of methanol, and the precipitated protein was removed by centrifugation at 17,000 g for 10 min. The supernatant was subjected to LC-MS analysis.

## Time-course analyses of ftChyM with 11 and 2

The reaction mixture (50 μL) containing 5 μM ftChyM, 200 μM compound **11**, 2.5 mM α-ketoglutaric acid, 0.4 mM FeSO$_4$, 4 mM vitamin C and 50 mM Tris-HCl (pH 7.5) in buffer C (pH 7.5) and was kept at 25 °C. The reaction was quenched by adding an equal volume of methanol after 0 min, 20 min, 1 h, 2 h, 4 h, 6 h and 8 h, respectively. The precipitated protein was removed by centrifugation at 17,000 g for 10 min. The supernatant was subjected to LC-MS analysis. Another reaction mixture (50 μL) containing 80 μM ftChyM, 200 μM

compound **2**, 2.5 mM α-ketoglutaric acid, 0.4 mM FeSO$_4$, 4 mM vitamin C and 50 mM Tris-HCl (pH 7.5) in buffer C (pH 7.5) and was kept at 25 °C. The reaction was quenched by adding 300 μL EtOAc after 5 min, 20 min, 40 min, 1 h and 2 h, respectively. The organic layer was dried *in vacuo*. The residue was dissolved in 100 μL methanol for LC-MS analysis.

### In vitro characterization of ftChyH

The plasmid pIM 3229 was transformed into the heterologous expression host *S. cerevisiae* BJ5464-NpgA through a Frozen-EZ Yeast Transformation II Kit (Zymo Research). The transformant yeast strains *BJ-ftchyH* were selected on solid selective uracil dropout medium at 30 °C for 2–3 days and confirmed by PCR. The right single colony was inoculated into 3 mL liquid uracil dropout medium and cultured at 30 °C and 250 rpm overnight. The yeast solution was inoculated on YPD liquid medium (20 g/L glucose, 20 g/L tryptone and 10 g/L yeast extract) and cultured at 28 °C and 250 rpm for 48 h. The culture broth was centrifuged to remove the solution and collect the cells. The cells were lysed by grinding, and cellular debris was resuspended in buffer C (pH 7.5). The cell-free extracts were harvested by centrifugation at 4 °C and 18,620 g for 30 min. To analyse the ftChyH-catalysed dehydrogenation reaction with substrate **5**, 100 μM **5** and 200 μM FAD were added to 100 μL ftChyH cell-free extracts to perform the reaction at 25 °C overnight. To show that ftChyH does not take part in catalysing the oxidative deamination of **2**, 400 μM **2** and 400 μM FAD were added to 100 μL ftChyH cell-free extracts to perform the reaction at 25 °C for 4 h. The reactions were quenched by adding 300 μL EtoAc and centrifuged at 17,000 g for 5 min to remove the precipitated proteins. The organic layer was dried *in vacuo*. The residue was dissolved in 100 μL methanol for LC-MS analysis. To show that ftChyH does not take part in catalysing the C-N bond oxidative cleavage of **11**, 100 μM **11** and 200 μM FAD were added to 100 μL ftChyH cell-free extracts to perform the reaction at 25 °C overnight. The reaction mixture was freeze-dried, and the residue was dissolved in 100 μL methanol for LC-MS analysis. The control assays were performed with yeast cell-free extract.

### ATP-PPi release analyses of ftChyA-A$_1$ and ftChyA-A$_2$

The substrate specificity of the ftChyA-A$_1$ and ftChyA-A$_2$ domains was characterized by ATP-PPi release. The EnzChek Pyrophosphate Assay Kit (E-6645, Molecular Probes) can be used for the quantitation of PPi in solution. The standard 100 μL reaction mixture containing 2 μM ftChyA-A$_1$ or ftChyA-A$_2$, 0.2 mM MESG substrate, 1 U purine nucleoside phosphorylase, and 0.03 U inorganic pyrophosphatase in 1×reaction buffer. After the reaction was preincubated at 22 °C for 10 min, it was initiated by the addition of 2 mM ATP, 20 mM MgCl$_2$·6H$_2$O, 1 mM substrate (L-Glu, L-Ala and Ant) for ftChyA-A$_1$, respectively. The reaction was performed with 2 mM ATP, 20 mM MgCl$_2$·6H$_2$O, 1 mM substrate (L-Ala and Ant) for ftChyA-A$_2$, respectively. The reaction proceeded at 22 °C for 30 min. The absorbance at 360 nm was then measured. A standard curve for the pyrophosphate assay can be generated using the pyrophosphate standard as a source of PPi. For data analysis, the values determined for the no-enzyme control were subtracted from the corresponding values of the experimental reaction. Data are shown as the mean ± SEM for 3 independent experiments, and the statistical analysis was calculated by unpaired two-tailed Student's *t* test using GraphPad Prism 7.

### Time-course analyses of the nonenzymatic reactions in Tris-HCl buffer

To show that compound **12** spontaneously converted to **4** and **6**, the reaction mixture (50 μL) containing 5 μM ftChyM, 200 μM **11**, 2.5 mM α-ketoglutaric acid, 0.4 mM FeSO$_4$, 4 mM vitamin C and 50 mM Tris-HCl (pH 7.5) in buffer C (pH 7.5) and was kept at 25 °C for 20 min. The reaction was quenched by adding 50 μL methanol and centrifuged at 17,000 g for 10 min. The products were analysed by

LC-MS after 0 min, 20 min, 1 h, 2 h, 4 h and 8 h at room temperature, respectively. To show that ftChyM does not take part in the cyclization of **4** to **6**, the assays were performed in 50 μL buffer C (pH 7.5) containing 20 μM ftChyM, 100 μM **4**, 2.5 mM α-ketoglutaric acid, 0.4 mM FeSO$_4$, 4 mM vitamin C and 50 mM Tris-HCl (pH 7.5) at 25 °C. The reactions were quenched after 5 min, 10 min, 20 min, 60 min and 120 min, respectively. The control assay was performed without ftChyM. To show that the α-carbonyl group in **4** greatly promotes spontaneous cyclization, the assays were performed in 50 μL buffer C (pH 7.5) containing 100 μM substrate (compounds **2**, **4** or **5**) at 25 °C. The reactions were quenched after 20 min, 1 h, 2 h, 4 h, 6 h and 8 h, respectively. To show that the alkaline-induced spontaneous C-2-N-3 bond closure of **4**, 100 μM **4** was added to 50 μL Tris-HCl buffer with different pH values (6.0, 7.5 and 9.0) at 25 °C. The reactions were quenched after 5 min, 10 min, 20 min and 60 min, respectively. All reactions were quenched by adding an equal volume of methanol and centrifuged at 17,000 g for 10 min before LC-MS analysis.

### Time-course analyses of the stability of 10 in Tris-HCl buffer or with ftChyA

To show that compound **10** does not spontaneously convert to **3**, 50 μL Tris-HCl buffer (pH 7.5) containing 200 μM **10** was incubated at 25 °C. To show that **3** is not derived from ftChyA with **10**, the 5 μM *holo* form of ftChyA was incubated with 200 μM **10** in 50 μL Tris-HCl buffer (pH 7.5) at 25 °C. The reactions were quenched after 1 h, 2 h, 4 h, 8 h and 12 h, respectively. All reactions were quenched by adding an equal volume of methanol and centrifuged at 17,000 g for 10 min before LC-MS analysis.

Other methods and results are available in the Supplementary information.

### Reporting summary

Further information on research design is available in the Nature Research Reporting Summary linked to this article.

## Data availability

We declare that all the data generated in this study are available within the main text and the Supplementary Information file. The sequence of *ftchy* gene cluster is provided in Source data or under the accession number OP651004 from NCBI: https://www.ncbi.nlm.nih.gov/. Plasmids, NCBI accession codes and hyperlinks for each protein used in this study can be found in Source data. Source data are provided as a Source Data file. Data is also available from the corresponding author upon request. Source data are provided with this paper.

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

## Acknowledgements

We thank Prof. Shi-Hui Dong from Lanzhou University and Prof. Zhao-Ming Dong from Southwest University for their help on MALDI-TOF MS analysis. This work is supported by the National Key R&D Programme of China (2020YFA0907700 to Y.Z.), Chongqing Science Funds for

Distinguished Young Scientists (cstc2020jcyj-jqX0005 to Y.Z.) and the 2035 pilot plan for innovative research of Southwest University (SWU-XDPY22009 to Y.Z.).

## Author contributions

X-W.C. performed all in vivo and in vitro experiments, as well as compounds isolation and characterization. L.R., J-L.C. performed compounds isolation and characterization. All authors analysed and discussed the results. Y.Z. supervised the research and wrote the manuscript.

## Competing interests

The authors declare no competing interests.
