## [Peer Review File · Nature Communications]

Unexpected assembly machinery for 4(3H)-quinazolinone scaffold synthesisREVIEWER COMMENTS

Reviewer #1 (Remarks to the Author):

The manuscript by Zou and coworkers describes an alternative biosynthetic pathway to the 4(3H)-quinazolinone core structure in fungi. The authors thoroughly investigate the chrysoquine biosynthetic pathway *in vivo* (e.g. recombinant, functional expression of the full pathway and of the NRPS system alone) and *in vitro* (in-depth studies on NRPS and tailoring enzymes) and put together a very conclusive picture on the entire biosynthesis of chrysoquine. All work appears to be planned and carried out with care, including necessary controls, all of which nicely represented in illustrative graphics throughout the manuscript. The work builds on previous investigations by Viggiano et al. (Appl. Environ. Microbiol. 2018) and Wollenberg et al. (J. Nat. Prod. 2017), who already described a dimodular NRPS system to be responsible for biosynthesis of chrysoquine, along with initial information on other biosynthetic aspects. The most important finding of the current study is the unusual catalytic activity of the NRPS, which catalyzes a 'reverse extension' in which the anthranilic acid residue bound to the first module serves as the N-nucleophile to attack the PCP-bound alanine residue on module 2 of the NRPS. This sequence is catalyzed by a terminal condensation domain, which likewise promotes subsequent hydrolytic cleavage of the product from the NRPS to give linear precursor which is further processed by a set of tailoring enzymes to give the final product. The novelty of this reported transformation is unfortunately a bit reduced due to the recent discovery of the Moore lab of a similar mechanism, termed pass-back mechanism, in thalassospiramide biosynthesis (Nat. Chem. Biol. 2020). While Moore examined this mechanism by *in vivo* approaches, the work now provided by the Zou group adds solid experimental evidence on the mechanism based on a multitude of suitable additional *in vitro* experiments. The investigations on the tailoring enzymes are likewise very detailed, including an interesting ftChyD-catalyzed amidation reaction as well as an ftChyM-catalyzed oxidative side-chain deamination reaction. In conclusion, while the work is thus based on quite some previous insights from other researcher on the same or mechanistically related systems, the extensive data presented here on this small but unique biosynthetic pathway deserves publication in a renowned journal such as Nature Communications, although a more specialized journal might be an even better fit.

The only request I would have is to carefully revise the manuscript text prior to resubmission of this work. There are quite some typos, very long sentences, etc., throughout the manuscript.

Reviewer #2 (Remarks to the Author):

The paper by Chen et. al. describes numerous interesting facets in the biosynthesis of 4(3H)-quinazolinones, focusing on a previously uncharacterized gene cluster from *Fusarium tricinctum* (ftchy). Individual domains from the biosynthetic pathway were characterized and compared to known domains. Initial findings showed that an uncommon 'pass-back' mechanism is present in a two-module NRPS (ftChyA) where reverse extension occurs in NRPS biosynthesis. While this has been observed before through *in vivo* mutations by Zhang et. al. in α -proteobacteria, this is the first time this mechanism has been confirmed in fungi and also the first time it has been reconstituted *in vitro*. An unusual amidation mechanism was also discovered whereby L-glutamine or inorganic ammonium was used to install the amide at N3 through ftChyD, a protein from the asparagine synthase superfamily, this is a novel mechanism of N3 installation in 4(3)-quinazolinone.

The authors have identified, purified and tested mechanistically many features of the ftChy biosynthetic pathway. A huge amount of work has been undertaken which must be congratulated and this report will certainly pave the way for further research into the biosynthesis of 4(3H)-quinazolinones as a very important structural motif present in many pharmaceutically important compounds.

However, while the data looks scientifically solid, there are a few important pieces of information missing in either the figures, methods or supplementary information. Before the manuscript can therefore be accepted in Nature Communications, the following minor yet crucial revisions need to be

implemented.

General Points:

- In Figure 4a, the pass-back mechanism was proven through in vitro assays with comparison to a synthetic standard of 3'. However, there is no trace of the synthetic standard of 3. To avoid speculation of whether this is the correct compound, it would be advisable to edit this figure to include synthetic standards of both 3 and 3', as 3 has been fully characterized in the supplementary information (supplementary table 10, supplementary figure 25, 39 and 40) and used in future assays (figure 6d). Whilst there is mass spectroscopy data for 3' (supplementary table 7, supplementary figure 34), there is no NMR data included in the supplementary information. Furthermore, while in the text it mentions that 3' was used as a comparison and a synthesized standard, it is not mentioned how this was synthesized or procured. I would suggest this data is added or an explanation of how 3' was synthesized/procured added as this distinction between the two products (3 and 3') is vital for the confirmation of the in vitro reconstitution of the 'pass-back' mechanism.

- In figure 6b there is LC-MS analysis of the incorporation of ¹⁵N into compound 2. While there is clear evidence that a mass shift of 1Da is observed, concurrent with the incorporation of a ¹⁵N isotope, it would be useful to add the spectrum of the mass of the negative control where unlabelled L-glutamine was used (figure 6a, iii).

- In supplementary figure 14 where the results of this feeding experiment are shown in full, there are multiple peaks present around the key ions identified showing very small mass changes (~0.3 Da differences). Do the authors have a theory as to why there are these peaks present in the spectrum. Also, there is a continuity error in the description of ftChyA/ftchyA in the figure.

- The labelling of figure 1 is different to that of all the other figures with the description then the part identifier, i.e. xxxx (a) not (a) xxxx. I suggest this is corrected to be consistent with the other figure labels.

- In the description of the motif in ftChyA-CT I believe the sentence should be 'Pro instead of His' as opposed to 'His instead of Pro' as written. The final paragraph of the ftChyA biochemical assays section needs to be rewritten as it contains many grammatical errors.

- Whilst the phylogenetic analysis of C domains showed a sequence motif for PHxxxD within fungal NRPS C domains (figure 4d, supplementary figures 8 and 10) it would be interesting to see whether the CT domains located within the thalassospiramide cluster also contain this motif.

Grammar Points:

There are a few spelling mistakes littered throughout the document, I suggest a thorough spell check.

Reviewer #3 (Remarks to the Author):

The manuscript "An alternative assembly machinery for 4(3H)-quinazolinone framework biosynthesis" concerns the biosynthesis pathway of the yellow fungal pigment chrysogine. In particular, the biosynthesis of the pharmacologically-relevant 4(3H)-quinazolinone scaffold and how its biosynthesis pathway differs from other 4(3H)-quinazolinone-containing natural products.

The authors first identify a biosynthetic gene cluster (ftchy) highly homologous to known chrysogine (1) in the genome of *Fusarium tricinctum*. *Fusarium tricinctum* is shown to be a producer of 1 and ftchy the gene cluster is confirmed by gene deletion. Aside from 1, several other related compounds are also produced (4-9), with 4,5, and 6 being on-pathway intermediates (confirmed through feeding studies on the ftchy-null mutant). 6 was shown to be converted into 1 via a reduction reaction catalysed by the SDR ftChyC, likely the final step in chrysogine biosynthesis.

Next, the activity of ftChyA (a bimoldular NRPS) is characterised in vitro. Incubating purified holo-ftChyA with anthranilic acid (Ant) and L-alanine resulted in the production of 3, the previously characterised product of the orthologous NRPS from *Penicillium chrysogenum*. Analysing the amino acid sequence of the two adenylation (A) domains of ftChyA indicated that, based on the predicted A domain selectivity residues, A1 should be selective for Ant while A2 is selective for L-ala. These substrate predictions are in opposition to the structure of 3 that, using canonical NRPS assembly logic, should be assembled by A1 selective L-ala and A2 selecting Ant. To further investigate the selectivity of the two A domain, pyrophosphate detection assays were performed. The results of these assays were consistent with the bioinformatic predictions that A1 adenylates Ant while A2 adenylates ala. The conserved His residue in condensation domain 1 (C1) is mutated and demonstrate to not be essential for the biosynthesis of 3. This result, together with the measured A domains selectivity assays, led the authors to propose that the terminal C domain (a CT domain – typically only catalyses chain release) catalyses both a “reverse extension” reaction, followed by hydrolysis of the product tethered to T1. An in vitro assay using the purified CT domain demonstrated this domain is sufficient for catalysing hydrolysis of 3.

The remainder of the paper concerns the role of the enzymes ftChyD, ftChyM, and ftChyH in chrysogine biosynthesis. ftChyD is demonstrated to catalyse the amidation of 3 to form 2 using L-glutamine or ammonium ions as amino donor. The N in the amide that forms becomes the N-3 in the 4(3H)-quinazolinone scaffold. ftChyM, and α -KGD, is then demonstrated to catalyse the deamination of 2 to produce 4. ftChyM could also oxidise the off-pathway product 5 to form 4. 4 then undergoes a spontaneous (pH dependent) cyclisation to produce 6. 6 is then reduced by ftChyC to form 1. The enzyme ftChyH is proposed to exclusively convert 5 back into 4, as the spontaneous conversion of 5 into 1 is slower than 4 into 6.

Main comment:

The most noteworthy result presented in this work is the proposed “reverse extension” and hydrolysis reaction catalysed by the CT domain of ftChyA. Such a finding would be of wide interest to the chemists and biologists in the biosynthesis and natural product research community. However, given the novelty such a finding would represent, I do not think the experiments performed conclusively demonstrate this mechanism.

Figure 5 contains diagrams for the three possible formation mechanisms of 3 by ftChyA. Diagram 3a follows canonical NRPS biosynthetic logic. The authors argument against mechanism 3a is that the A domains of ftChyA appear to have the opposite selectivity than expected. One issue with the experimental approach to determine the selectivity of the A domains is that, as far as I can tell from the methods no acyl acceptor/quencher (such as hydroxylamine) is included in the assay. Without an acyl acceptor, the release of the adenylated substrate from the active site of the A domain is by leakage only, meaning that pyrophosphate levels can be higher for poorer/non-native substrate, as these are bound less tightly by the A domain (A nice discussion of A domain activity assay can be found here: A. Stanišić, H. Kries, *ChemBioChem* 2019, 20, 1347). As such, I am not convinced the A domain assays performed here are robust enough for the argument the authors are making.

The authors argue against mechanism 5b being unable to detect 3' in their assays. This is valid.

Next up is mechanism 3c (the proposed mechanism). Mutating the conserved His residues in the active site of C1 does not affect the production of 3, leading the authors to conclude that C1 is catalytically inactive and does not participate in the formation of 3. As there is only one C domain left (CT), it is argued that this C domain must catalyse both a “reverse extension” reaction of Ant and ala, and hydrolytic release of 3. There are several issues here. The first being that C1 is inactive because the conserved His residue is not essential. The catalytic mechanism of NRPS C domain is still unclear. While the conserved His is often proposed to play a role as a general base (or perhaps substrate positioning) to accelerate the condensation reaction, mutating it does not always abolish condensation activity (discussion and primary references can be found in this review: Bloudoff K, Schmeing TM. Structural and functional aspects of the nonribosomal peptide synthetase

condensation domain superfamily: discovery, dissection and diversity. *Biochim Biophys Acta Proteins Proteom.* 2017;1865(11 Pt B):1587-1604. doi:10.1016/j.bbapap.2017.05.010. As such, the fact that mutating the His of C1 did not abolish its activity does not rule out that it could still be catalysing a condensation reaction (meaning 5a is the true mechanism).

To prove that mechanism 5c is true the authors need to conclusively demonstrate the Ant-ala thioester bound to T1 really exists. One way to do this could be adding trypsin at various points during the in vitro assay then analysing the peptide fragments to find the predicted 3 product bound to the serine-containing fragment of T1. However, another possible mechanism is that, after the reverse extension, C1 transfers the substrate to T2 for hydrolytic offloading by the CT. This mechanism should also be considered, but would be difficult to differentiate from mechanism 5a.

Other comments:

The work with the enzymes ftChyD, ftChyM, and ftChyH seems, generally, solid. But unless the points raised above can be addressed I do not think they would merit being published alone in *Nature Communications*.

In the abstract/conclusion: Our study uncovers a novel function of NRPS C domains, shows new route for 4(3H)-quinazolinone, and supports its further application for the development of pharmaceuticals. Synthetic routes to this scaffold are already known and used. As such, how do these findings help pharmaceutical development? This should be better explained and justified if it is indeed the case.

I suggest a word other than “framework” is used in the title and throughout. I think scaffold would be better.

Do the homologous adenylation domains in ftChyA homologues also have the same selectivity motifs/activity? If so it would lend some support to the “reverse extension” mechanism.

The first enzyme functionally investigated (ftChyC) is the last one in the pathway. It would be easier to follow if the enzymes of the pathway were investigated in order.

One of the goals of the study is to find “the mechanism by which NRPS ChyA synthesizes 3 and its two modified products”. But the mechanism for these modified products is not described.

There are many LC-MS traces in the main results in this paper, many of which I think would be better suited in the supplementary. I would only keep the key traces demonstrating the activity of each studied enzyme. This would make the manuscript more streamlined and easier to follow.

The statement: “Although the previously identified on-pathway intermediate 2 was not directly detected in strain AN-ftchyACDEHM, the production of 7 highly indicates that small amounts of 2 should be converted by acetylation in *A. nidulans*”. It would have been good to feed 2 to the ftChyA mutant, just to confirm it really is an intermediate in this particular strain (I don't doubt it is though). But why should 7 necessarily be formed in *A. nidulans*? There is no evidence for this.

The statement: “These in vitro results (1) fully establish the solid relationship of ftChyA with 3; (2) demonstrate that the A domains of ftChyA recognize only anthranilic acid and L-alanine as substrates”. To establish this more substrates (such as the 20 common amino acids) should be tested to demonstrate that the A domains are the most active with Ant/ala compared to these other common substrates.

The statement: “These results demonstrate that N-3 within the 4(3H)-quinazolinone framework of 1 comes from the ϵ -amino group of L-glutamine or the inorganic ammonium ion, which is substantially different from the previous mechanisms shown in Fig. 1c and 1d.” This is very vague. The authors

should explicitly state exactly what they mean: how is it different?

The statement: "Further mutation of the conserved P2074 to A or H abolished the production of 3 by ftChyA (Fig. 4f, iv and v), which showed P2074 is also essential for ftChyA-CT." I do not think this result adds anything to the arguments made in the paper. The authors give no explanation for this result (which is almost certainly due to structural changes).

The statement: "Recently, via in vivo mutation investigations, Moore and colleagues observed this unusual chain reverse extension (namely pass-back mechanism) of NRPS is existed in assembly of thalassospiramide family compounds from α - proteobacteria³¹. Therefore, combination of our in vitro biochemical confirmation of fungal two- module NRPS ftChyA here, these two works open the door to explore the new biosynthetic logic of NRPSs, both in prokaryotes and eukaryotes, for the synthesis of peptide compounds." This should be moved to the discussion and the authors should spend more time explaining exactly what the Moore lab discovered and how it is similar or different to what they have discovered (which they claim to be novel). Without this the results of this work have no proper context.

At several points in the paper, reaction rates are compared and used to make arguments about the true nature of the biosynthesis pathway (such as the cyclisation rate between 4 and 6) (see Fig 6e and Supp Fig 19). However, no formal rate measurements were made, leading to vague and unscientific sounding statements such as "We noted that the spontaneous cyclization of 5 into 1 is much slower than that of 4 into 6", "The activity of ftChyM slightly decrease in the absence of exogenous ferrous ions (Fig. 6d, iii)", and "Although these cyclization products were observed (Supplementary Fig. 19), their conversion rates were far behind those of compounds 4 to 6...". Putting some actual numbers to the rates would made the arguments better.

Figure 6a: the contrast between the black and the dark blue is not strong enough, making the figure hard to follow. Also, the meaning of the X8 is not clear from the legend.

Typo: "...or dehydrogenation to from [form] 4, where the resultant α - carbonyl group in 4 greatly promotes final spontaneous cyclization to form the 4(3H)- quinazolinone framework."

Typo: "...an α -KGD [that] converts GA4 to GA7 by forming a 1,2 carbon double bond during gibberellic acid biosynthesis³⁷."

Typos: "Therefore, aprat [apart] from the extra ability of ftChyM-catalysed dehydrogenation of 5 to give 4, we proposed that ftChyH is also invovled [involved] in this transformation in strain AN-ftchyCDEHM."

Response to Reviewers' Comments

We thank the reviewers for their most insightful comments. We have addressed each comment carefully and a point-by-point reply is listed below. We believe the changes made according to their suggestions have significantly strengthened the manuscript.

Review 1 Comments	Reply/Changes
The only request the reviewer would have is to carefully revise the manuscript text prior to resubmission of this work. There are quite some typos, very long sentences, etc., throughout the manuscript.	Thank you. The manuscript has been polished by the professional services.
Review 2 Comments	Reply/Changes
In Figure 4a, the pass-back mechanism was proven through in vitro assays with comparison to a synthetic standard of 3'. However, there is no trace of the synthetic standard of 3. To avoid speculation of whether this is the correct compound, it would be advisable to edit this figure to include synthetic standards of both 3 and 3', as 3 has been fully characterized in the supplementary information (supplementary table 10, supplementary figure 25, 39 and 40) and used in future assays (figure 6d). Whilst there is mass spectroscopy data for 3' (supplementary table 7, supplementary figure 34), there is no NMR data included in the supplementary information. Furthermore, while in the text it mentions that 3' was used as a comparison and a synthesized standard, it is not mentioned how this was synthesized or procured. I would suggest this data is added or an explanation of how 3' was synthesized/procured added as this distinguishment between the two products (3 and 3') is vital for the confirmation of the in vitro reconstitution of the 'pass-back' mechanism.	Thank you for your helpful suggestion. Compound 3 was purified from the ftChyA in vitro assays, its structure was confirmed by NMR and HR-MS (supplementary Table 9 and Figs. 34, 53-57). Compound 3' is a commercial product of ChinaPeptides Co., Ltd, and its structure was also confirmed by us via NMR and HR-MS analyses. In the original manuscript, it was only used as a standard to compare and demonstrate that the product of ftChyA is indeed compound 3 but not 3'. To avoid misunderstanding, we deleted the 3' from the revised manuscript.
In figure 6b there is LC-MS analysis of the incorporation of 15N into compound 2. While there is clear evidence that a mass shift of 1Da is observed, concurrent with the incorporation of a 15N isotope, it would be useful to add the spectrum of the mass of the negative control where unlabelled L-glutamine was used (figure 6a, iii).	Thank you for your helpful suggestion. The negative control of unlabelled L-glutamine was added in new supplementary Fig. 17.
In supplementary figure 14 where the results of this feeding experiment are shown in full, there are multiple peaks present around the key ions identified showing very small mass changes (~0.3 Da differences). Do the authors have a theory as to why there are these peaks present in the spectrum. Also, there is a continuity error in the description of ftChyA/ftchyA in the figure.	Thank you. The multiple peaks present around the key ions are from the in vitro reaction system. To eliminate this interference, a small amount of product was purified from the in vitro assays to perform the HRMS analysis. The new supplementary Fig. 17 has been updated.
The labelling of figure 1 is different to that of all the other figures with the description then the part identifier, i.e. xxxx (a) not (a) xxxx. I suggest this is corrected to be consistent with the other figure labels.	Thank you for your helpful suggestion. Changed as suggested.
In the description of the motif in ftChyA-CT I believe the sentence should be 'Pro instead of His' as opposed to 'His instead of Pro' as written.	Thank you for your helpful suggestion. Changed as suggested.
The final paragraph of the ftChyA biochemical assays section needs to be rewritten as it contains many grammatical errors.	Thank you for your helpful suggestion. The ftChyA biochemical assays section was rewritten and the language was polished by the professional services.
Whilst the phylogenetic analysis of C domains showed a sequence motif for PHxxxD within fungal NRPS C domains (figure 4d, supplementary figures 8 and 10) it would be interesting to see whether the CT domains located within the thalassospiramide cluster also contain this motif.	Thank you for your helpful suggestion. The conserved motifs of AGC65516.1_C2 domain and WP_062953557.1_C2 domain of thalassospiramide cluster are DHxxxD.

There are a few spelling mistakes littered throughout the document, the reviewer suggests a thorough spell check.	Thank you for your helpful suggestion. The spelling mistakes have been corrected.
Review 3 Comments	Reply/Changes
The most noteworthy result presented in this work is the proposed “reverse extension” and hydrolysis reaction catalysed by the CT domain of ftChyA. Such a finding would be of wide interest to the chemists and biologists in the biosynthesis and natural product research community. However, given the novelty such a finding would represent, I do not think the experiments performed conclusively demonstrate this mechanism. Figure 5 contains diagrams for the three possible formation mechanisms of 3 by ftChyA. Diagram 3a follows canonical NRPS biosynthetic logic. The authors argument against mechanism 3a is that the A domains of ftChyA appear to have the opposite selectivity than expected. One issue with the experimental approach to determine the selectivity of the A domains is that, as far as I can tell from the methods no acyl acceptor/quencher (such as hydroxylamine) is included in the assay. Without an acyl acceptor, the release of the adenylated substrate from the active site of the A domain is by leakage only, meaning that pyrophosphate levels can be higher for poorer/non-native substrate, as these are bound less tightly by the A domain (A nice discussion of A domain activity assay can be found here: A. Stanišić, H. Kries, ChemBioChem 2019, 20, 1347). As such, I am not convinced the A domain assays performed here are robust enough for the argument the authors are making. The authors argue against mechanism 5b being unable to detect 3' in their assays. This is valid. Next up is mechanism 3c (the proposed mechanism). Mutating the conserved His residues in the active site of C1 does not affect the production of 3, leading the authors to conclude that C1 is catalytically inactive and does not participate in the formation of 3. As there is only one C domain left (CT), it is argued that this C domain must catalyse both a “reverse extension” reaction of Ant and ala, and hydrolytic release of 3. There are several issues here. The first being that C1 is inactive because the conserved His residue is not essential. The catalytic mechanism of NRPS C domain is still unclear. While the conserved His is often proposed to play a role as a general base (or perhaps substrate positioning) to accelerate the condensation reaction, mutating it does not always abolish condensation activity (discussion and primary references can be found in this review: Bloudoff K, Schmeing TM. Structural and functional aspects of the nonribosomal peptide synthetase condensation domain superfamily: discovery, dissection and diversity. Biochim Biophys Acta Proteins Proteom. 2017;1865(11 Pt B):1587-1604. doi:10.1016/j.bbapap.2017.05.010. As such, the fact that mutating the His of C1 did not abolish its activity does not rule out that it could still be catalysing a condensation reaction (meaning 5a is the true mechanism). To prove that mechanism 5c is true the authors need to conclusively demonstrate the Ant-ala thioester bound to T1 really exists. One way to do this could be adding trypsin at various points during the in vitro assay then analysing the peptide fragments to find the predicted 3 product bound to the serine-containing fragment of T1.	We deeply thank the reviewer’s suggestions. Here, we pay tribute to the reviewers' professional comments, which helps and guides us to reconsider the mechanism of the two-module NRPS ftChyA. According to the reviewer’s comments, we first investigated the function of ftchyA in vivo. To our surprise, a tripeptide 10 is discovered produced by AN-ftchyA only via MeOH extraction. When 1 mM Ant was fed, the dipeptide 3 was also detected. Therefore, we re-carried out a series of in vitro assays of ftChyA (with three substrates, L-Glu, L-Ala and Ant), which confirmed that (1) the tripeptide 10 is indeed the main product of ftChyA and the dipeptide 3 is the shunt product of ftChyA (without L-Glu); (2) the C_T domain is indispensable to produce both 10 and 3, however, the C₁ domain is only essential to produce 10; (3) via the MALDI-TOF MS analysis of the trypsin-digested T domain and the ATP-PPi release assays, we confirmed that the A₁ domain recognizes L-Glu and loads it to the T₁ domain and A₂ domain recognizes L-Ala. Based on these new results, we proposed three mechanisms of ftChyA to produce 10 and 3 (Figure 5, mechanisms a-c). With the further assistance of experiments with H₂¹⁸O and anthranilate methyl ester (Ant-Me), we excluded the mechanism a and b, and finally demonstrated that the mechanism c is possible the actual program rule of ftChyA to synthesize 10 and 3. The new mechanism of ftChyA represents an expected assembly machinery of fungal two-module NRPS to synthesize linear tripeptide, where an unusual C_T domain catalyzes release of the on-line ε-L-glutamyl-L-alanyl-S-T₂ or L-alanyl-S-T₂ via the off-line anthranilate. We deeply thank the reviewer again for promoting us to re-investigate the mechanism of ftChyA.

However, another possible mechanism is that, after the reverse extension, C1 transfers the substrate to T2 for hydrolytic offloading by the CT. This mechanism should also be considered, but would be difficult to differentiate from mechanism 5a.	
The work with the enzymes ftChyD, ftChyM, and ftChyH seems, generally, solid. But unless the points raised above can be addressed I do not think they would merit being published alone in Nature Communications.	Thank you. We confirmed the function of ftChyD, ftChyM, ftChyE and ftChyH, respectively, and showed that they-catalyzed pathway from 3 to 1 is the salvage route for the synthesis of 1 (Figure 7).
In the abstract/conclusion: Our study uncovers a novel function of NRPS C domains, shows new route for 4(3H)-quinazolinone, and supports its further application for the development of pharmaceuticals. Synthetic routes to this scaffold are already known and used. As such, how do these finding help pharmaceutical development? This should be better explained and justified if it is indeed the case.	Thank you for your helpful suggestion. We adjusted the conclusion part of the abstract: Our study uncovers a unique release and tailoring mechanism of nonribosomal peptides and a new route for the synthesis of 4(3H)-quinazolinone scaffolds.
The reviewer suggests a word other than “framework” is used in the title and throughout. The reviewer think scaffold would be better.	Thank you for your helpful suggestion. Changed as suggested.
Do the homologous adenylation domains in ftChyA homologous also have the same selectivity motifs/activity? If so it would lend some support to the “reverse extension” mechanism.	Thank you. Sequence alignment of ftChyA-A ₁ and ftChyA-A ₂ with its homologous enzymes showed that 75% and 77% identity to homologous enzymes, respectively.
The first enzyme functionally investigated (ftChyC) is the last one in the pathway. It would be easier to follow if the enzymes of the pathway were investigated in order.	Thank you for your helpful suggestion. The ftChyC-catalyzed reduction is the last step during 1 synthesis. We confirmed its function firstly due to the compounds isolated from the heterologous strain AN-ftchyACDEHM , thus we wrote the section of ftChyC following the heterologous part.
One of the goals is to find “the mechanism by which NRPS ChyA synthesizes 3 and its two modified products”. But the mechanism for these modified products is not described.	Thank you. These two modified products are possibly modified by the unknown enzymes (out of the chy cluster) of the host P. chrysogenum (ref 18), because these two compounds were not discovered in F. graminearum (ref 19), and F. tricinctum of this work.
There are many LC-MS traces in the main results in this paper, many of which I think would be better suited in the supplementary. I would only keep the key traces demonstrating the activity of each studied enzyme. This would make the manuscript more streamlined and easier to follow.	Thank you for your helpful suggestion. We adjusted the figures in main text, where a lot of LC-MS traces were moved into the supplementary information.
The statement: “Although the previously identified on-pathway intermediate 2 was not directly detected in strain AN-ftchyACDEHM, the production of 7 highly indicates that small amounts of 2 should be converted by acetylation in A. nidulans ”. It would have been good to feed 2 to the ftChyA mutant, just to confirm it really is an intermediate in this particular strain (I don’t doubt it is though). But why should 7 necessarily be formed in A. nidulans ? There is no evidence for this.	Thank you. Production of 7 in AN-ftchyACDEHM is possibly due to an unknown N-acetyltransferase of the heterologous host A. nidulans . We proved this hypothesis by feeding 2 into A. nidulans , the production of 7 was observed (supplementary Fig. 31). Strain A. nidulans indeed has an ability to take N-acetylation on product, for example, pyrophen (J Nat Prod 2020, 83, 593-600). Therefore, we concluded that the production of 7 is due to the over-modification of 2 by unknown acetyltransferase of A. nidulans .
The statement: “These in vitro results (1) fully establish the solid relationship of ftChyA with 3 ; (2) demonstrate that the A domains of ftChyA recognize only anthranilic acid and L-alanine as substrates”. To establish this more substrates (such as the 20 common amino acids) should be tested to demonstrate that the A domains are the most active with Ant/ala compared to these other common substrates.	Thank you. We now proved that the A domains of ftChyA recognize L-Glu and L-Ala, respectively (Fig. 4d).
The statement: “These results demonstrate that N-3 within the 4(3H)-quinazolinone framework of 1 comes from the ε- amino group of L-glutamine or the inorganic	Thank you. The original source of N-3 of the 4(3H)-quinazolinone scaffold in 1 is one of the main issues in this work. The inorganic ammonium ion, or the ε-NH ₂

ammonium ion, which is substantially different from the previous mechanisms shown in Fig. 1c and 1d.” This is very vague. The authors should explicitly state exactly what they mean: how is it different?	of L-Gln was demonstrated to be incorporated into the N-3 by labeled substrates and biochemical assays. It is a new source for the 4(3H)-quinazolinone scaffold, because the previous mechanisms (Fig. 1c and 1d) showed that the N-3 of 4(3H)-quinazolinone scaffold comes from the α -NH ₂ of amino acids or their analogs (building blocks).
The statement: “Further mutation of the conserved P2074 to A or H abolished the production of 3 by ftChyA (Fig. 4f, iv and v), which showed P2074 is also essential for ftChyA-CT.” I do not think this result adds anything to the arguments made in the paper. The authors give no explanation for this result (which is almost certainly due to structural changes).	Thank you. We agree with your comment. The real role of P ₂₀₇₄ in ftChyA is indeed not clarified only by the current mutation results. Therefore, we removed these results and discussion in revised manuscript.
The statement: “Recently, via in vivo mutation investigations, Moore and colleagues observed this unusual chain reverse extension (namely pass-back mechanism) of NRPS is existed in assembly of thalassospiramide family compounds from α -proteobacteria. Therefore, combination of our in vitro biochemical confirmation of fungal two-module NRPS ftChyA here, these two works open the door to explore the new biosynthetic logic of NRPSs, both in prokaryotes and eukaryotes, for the synthesis of peptide compounds.” This should be moved to the discussion and the authors should spend more time explaining exactly what the Moore lab discovered and how it is similar or different to what they have discovered (which they claim to be novel). Without this the results of this work have no proper context.	Thank you. We now demonstrated that the two-module NRPS ftChyA synthesizes a linear tripeptide 10 and dipeptide 3 via release of the on-line ϵ -L-glutamyl-L-alanyl-S-T ₂ or L-alanyl-S-T ₂ using the off-line anthranilate, respectively.
At several points in the paper, reaction rates are compared and used to make arguments about the true nature of the biosynthesis pathway (such as the cyclisation rate between 4 and 6) (see Fig 6e and Supp Fig 19). However, no formal rate measurements were made, leading to vague and unscientific sounding statements such as “We noted that the spontaneous cyclization of 5 into 1 is much slower than that of 4 into 6”, “The activity of ftChyM slightly decrease in the absence of exogenous ferrous ions (Fig. 6d, iii)”, and “Although these cyclization products were observed (Supplementary Fig. 19), their conversion rates were far behind those of compounds 4 to 6...”. Putting some actual numbers to the rates would made the arguments better.	Thank you for your helpful suggestions. We adjusted our descriptions on these conclusions in the revised manuscript.
Figure 6a: the contrast between the black and the dark blue is not strong enough, making the figure hard to follow. Also, the meaning of the X8 is not clear from the legend.	Thank you for your helpful suggestion. Changed as suggested.
Typo: “...or dehydrogenation to from [form] 4, where the resultant α - carbonyl group in 4 greatly promotes final spontaneous cyclization to form the 4(3H)- quinazolinone framework.”	Thank you. Changed as suggested.
Typo: “...an α -KGD [that] converts GA4 to GA7 by forming a 1,2 carbon double bond during gibberellic acid biosynthesis.”	Thank you. Changed as suggested.
Typos: “Therefore, aprat [apart] from the extra ability of ftChyM-catalysed dehydrogenation of 5 to give 4, we proposed that ftChyH is also invovled [involved] in this transformation in strain AN- ftchyCDEHM.	Thank you. Changed as suggested.

REVIEWER COMMENTS

Reviewer #2 (Remarks to the Author):

The authors have addressed all our comments and questions in a satisfactory manner. I therefore support acceptance of this manuscript.

Reviewer #3 (Remarks to the Author):

This manuscript has undergone significant changes since its first submission. In the first submission the enzyme ftChyA was proposed to synthesize the dipeptide 3 via a "pass-back" mechanism, which was a standout finding from the report. In the revised submission this "pass back" mechanism is no longer proposed due to the identification of the tripeptide 10, the likely true product of ftChyA. The authors demonstrate that the C-terminal CT domain of ftChyA is responsible for catalysing the chain release of a Glu-Ala dipeptide using the amine of L-Ant as a nucleophile. The rest of the manuscript concerns the function of the tailoring enzymes ftChyE, ftChyM, ftChyE and their activity with 10, or derivatives of 10. As such, the biochemical pathway proposed for the biosynthesis of the 4(3H)-quinazolinone scaffold of 1 is very different.

This paper represents a huge amount of work using a diverse range of techniques. The paper includes compound isolation, structural characterisation, gene knockouts, heterologous expression, and protein purification. Several of the experiments proved to be difficult but the authors managed to successfully troubleshoot (such as but changing the heterologous host, or by using a different protein fusion tag). The authors should be commended. I support publication but think the authors should be careful with some of their conclusions.

Some comments

Synthesising 1 via 10 is, as the authors note, unexpected. It would be good if they expanded upon this in the discussion, given that it seems completely unnecessary to incorporate Glu only for it to then be removed. Why is this so unexpected? I feel this is not properly addressed. It is possible the tripeptide is converted in vivo to a different natural product, and 1 is the shunt product?

I disagree that the terminal CT domain is really all that novel. CT domains that catalyse chain release by selecting an exogenous amine have been identified (mentioned in reference 34). The only difference here is that an amino acid is selected, the general mechanism has been identified previously.

While the manuscript is perfectly readable, there are numerous typos/incorrect wordings/tenses that stand out. Editing by an English-native proof reader with a scientific background would improve the manuscript.

Response to Reviewers' Comments

We thank the reviewers for their most insightful comments. We have addressed each comment carefully and a point-by-point reply is listed below. We believe the changes made according to their suggestions have significantly strengthened the manuscript.

Review 3 Comments	Reply/Changes
This paper represents a huge amount of work using a diverse range of techniques. The paper includes compound isolation, structural characterisation, gene knockouts, heterologous expression, and protein purification. Several of the experiments proved to be difficult but the authors managed to successfully troubleshoot (such as but changing the heterologous host, or by using a different protein fusion tag). The authors should be commended. I support publication but think the authors should be careful with some of their conclusions.	Thank you.
Synthesising 1 via 10 is, as the authors note, unexpected. It would be good if they expanded upon this in the discussion, given that it seems completely unnecessary to incorporate Glu only for it to then be removed. Why is this so unexpected? I feel this is not properly addressed. It is possible the tripeptide is converted in vivo to a different natural product, and 1 is the shunt product?	Thank you for your helpful suggestions. We added a paragraph to discuss the unexpected synthetic pathway from 10 to 1. “As shown in Fig. 7, synthesis of 1 from 10 represents a new route for generating the 4(3H)-quinazolinone scaffold. Apart from the inorganic substrates are involved in 1 synthesis (NH_4^+ for N-3 and water for C-1'-OH), the main unexpected assembly machinery is that L-Glu is first recruited by ftChyA to synthesize 10, however, it is then removed as L-Gln by ftChyM (via oxidative cleavage) during the post tailoring steps of 1. This seems a redundant process for the formation of 1; however, it is particularly worth mentioning that the generated L-Gln from ftChyM reaction could be recaptured and hydrolysed by ftChyD to yield L-Glu and ammonium ions, where these two products could re-participate in ftChyA-catalysed 10 formation and ftChyD-catalysed amidation reactions, respectively. Therefore, from this perspective, an efficient self-circulation system among ftChyA, ftChyD and ftChyM-catalysed reactions has been established during the synthetic process of 1 (Fig. R1, overleaf).” We further searched the SciFinder and Reaxys database, there are no natural products containing 10 or 11 as the structural building block. Moreover, apart from 2, 3 and 4, we do not observe other compounds derived from 10 or 11 during our experiments. Thus, 1 should be the product of the ftchy cluster, but not the shunt product. We deeply thank reviewer's professional comments on this point, this helps us better understand the synthetic logic of 1.
I disagree that the terminal C_T domain is really all that novel. C_T domains that catalyse chain release by selecting an exogenous amine have been identified (mentioned in reference 34). The only difference here is that an amino acid is selected, the general mechanism has been identified previously.	Thank you for your helpful suggestions. We rewrote the sentences about the conclusion of ftChyA C_T domain as “These results demonstrate that ftChyA-C_T is a unique C domain, representing an unusual function of fungal two-module NRPS C domains.”
While the manuscript is perfectly readable, there are numerous typos/incorrect wordings/tenses that stand out. Editing by an English-native proofreader with a scientific background would improve the manuscript.	Thank you for your helpful suggestions. We have used the English Language Editing Service of American Journal Experts (partner of SPRINGER NATURE) to polish the languages of the whole manuscript.

Fig. R1 An efficient self-circulation system among ftChyA, ftChyD and ftChyM-catalysed reactions. (Supplementary Fig. 84 in SI).

REVIEWERS' COMMENTS

Reviewer #3 (Remarks to the Author):

The authors have addressed the concerns. I support publication of this manuscript.